# Novel mechanisms of MITF regulation identified in a mouse suppressor screen

Hong Nhung Vu [1], Matti Már Valdimarsson[2], Sara Sigurbjörnsdóttir [1], Kristín Bergsteinsdóttir[1], Julien Debbache[3], Keren Bismuth[3], Deborah A Swing[4], Jón H Hallsson [1], Lionel Larue [5], Heinz Arnheiter [3], Neal G Copeland[4,6], Nancy A Jenkins[4,6], Petur O Heidarsson [2] & Eiríkur Steingrímsson [1]✉

## Abstract

**MITF, a basic Helix-Loop-Helix Zipper (bHLHZip) transcription factor, plays vital roles in melanocyte development and functions as an oncogene. We perform a genetic screen for suppressors of the Mitf-associated pigmentation phenotype in mice and identify an intragenic Mitf mutation that terminates MITF at the K316 SUMOylation site, leading to loss of the C-end intrinsically disordered region (IDR). The resulting protein is more nuclear but less stable than wild-type MITF and retains DNA-binding ability. As a dimer, it can translocate wild-type and mutant MITF partners into the nucleus, improving its own stability thus ensuring nuclear MITF supply. smFRET analysis shows interactions between K316 SUMOylation and S409 phosphorylation sites across monomers; these interactions largely explain the observed effects. The recurrent melanoma-associated E318K mutation in MITF, which affects K316 SUMOylation, also alters protein regulation in concert with S409. This suggests that residues K316 and S409 of MITF are impacted by SUMOylation and phosphorylation, respectively, mediating effects on nuclear localization and stability through conformational changes. Our work provides a novel mechanism of genetic suppression, and an example of how apparently deleterious mutations lead to normal phenotypes.**

**Keywords** *Mitf*; Protein Stability; Nuclear Export; Suppressor; Transcription
**Subject Categories** Cancer; Post-translational Modifications & Proteolysis; Skin

## Introduction

Transcription factors play a crucial role in gene regulation, and most of them have large unstructured domains termed intrinsically disordered regions (IDRs) in addition to their DNA-binding domains (Már et al, 2023). Due to the lack of tools and links to phenotypes, understanding the structure-function relationships of IDRs and their specific contributions to in vivo activity and disease has proven challenging. The basic Helix-Loop-Helix-leucine zipper (bHLHZip) transcription factor MITF is the master regulator of melanocyte development and pigmentation. It also plays a critical role in melanoma, a highly aggressive skin cancer originating from melanocytes (Goding and Arnheiter, 2019; Rambow et al, 2019). Importantly, MITF protein activity can be modulated either transiently through environmental signals or permanently by mutations leading to critical effects on the phenotype. In melanoma, MITF activity mediates phenotype plasticity such that high MITF activity promotes differentiation and proliferation, whereas low MITF activity results in a stem cell-like phenotype and enhances migration (Rambow et al, 2019). MITF binds to E-(CACGTG) and M- (TCATGTG) box motifs as a homodimer or as a heterodimer with its closest relatives, TFE3, TFEB, and TFEC (Hemesath et al, 1994; Laurette et al, 2015). A unique 3-amino acid sequence in the zipper domain restricts dimerization of these proteins such that they do not dimerize with other bHLHZip proteins (Liu et al, 2023; Pogenberg et al, 2020; Pogenberg et al, 2012). Outside the bHLHZip domain, MITF consists of N- and C-terminal IDRs, located on either side of the bHLHZip DNA-binding and dimerization domain, largely of unknown function.

Importantly, multiple phosphorylation sites have been mapped in the IDRs of MITF (Vu et al, 2021), and some (including S69, S73, and S173) have been suggested to lead to nuclear export or retention of MITF in the cytoplasm, some (S73 and S409) to affect transcription regulation and other sites have been proposed to affect protein stability (S73, S397, S401, S405, and S409) (Fig. 1A); (Vu et al, 2021). Interestingly, the S73 and S409 residues have been shown to be priming sites for GSK3β-mediated phosphorylation of

[1]Department of Biochemistry and Molecular Biology, BioMedical Center, Faculty of Medicine, University of Iceland, Sturlugata 8, 102 Reykjavík, Iceland. [2]Department of Biochemistry, Science Institute, School of Engineering and Natural Sciences, University of Iceland, Sturlugata 7, 102 Reykjavík, Iceland. [3]Mammalian Development Section, NINDS, NIH, Bethesda, MD 20892-3706, USA. [4]Mouse Cancer Genetics Program, NCI, Frederick, MD 21702-1201, USA. [5]Institut Curie, PSL Research University, INSERM U1021, Normal and Pathological Development of Melanocytes, 91405 Orsay, France. [6]Present address: Genetics Department, MD Anderson Cancer Center, Houston, TX 77030, USA. ✉E-mail: eirikurs@hi.is

    

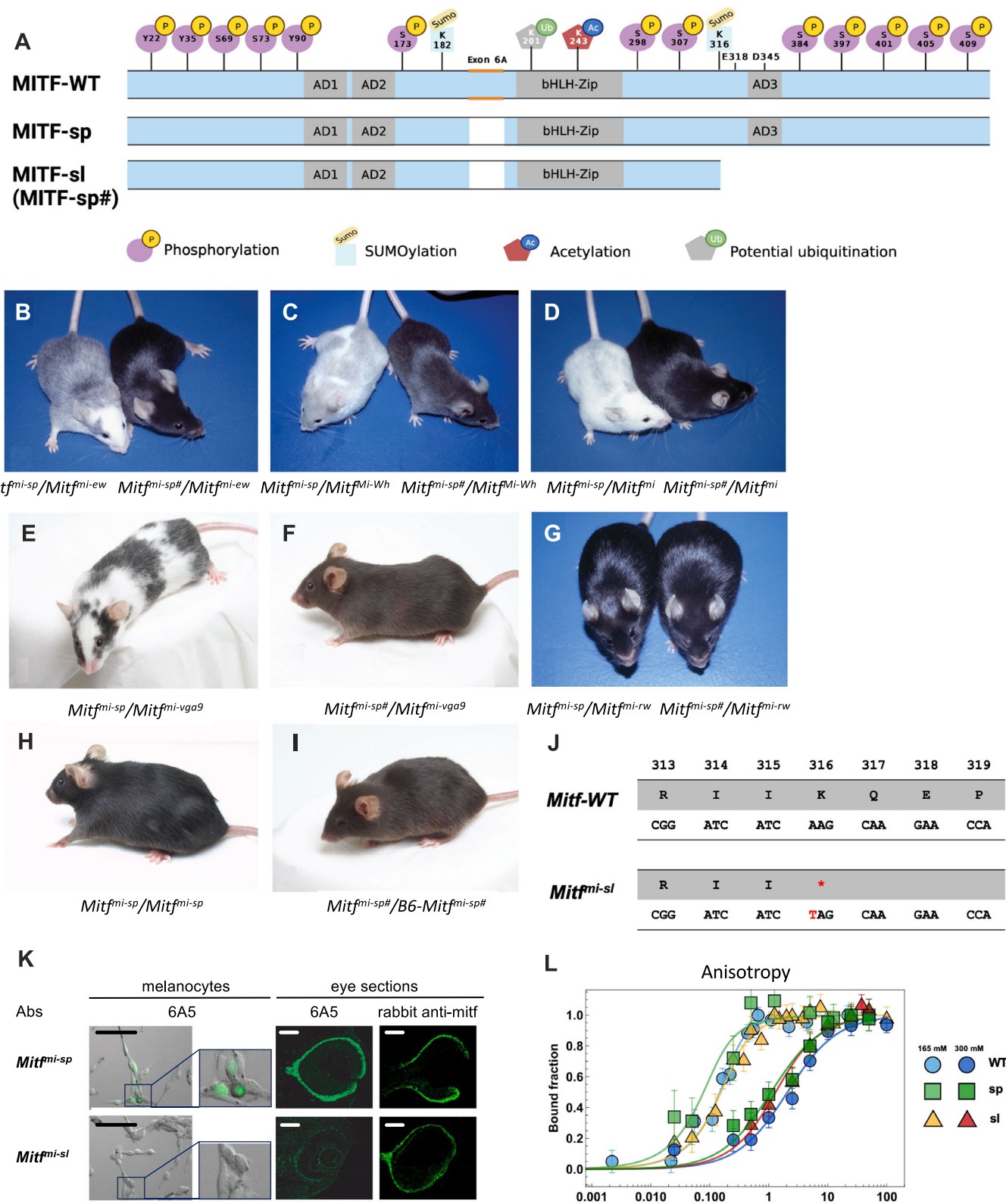

**Figure 1. Coat color phenotypes and molecular alteration associated with the induced *Mitf^mi-sp#* suppressor mutation (*Mitf^mi-sl*).**

(A) Graphical depiction of the MITF-WT, MITF-sp, and MITF-sl (MITF-sp#) proteins indicating the domains affected. Also shown are the post-translational modifications that have been reported in MITF. (B) NAW-*Mitf^mi-ew*/B6-*Mitf^mi-sp* and NAW-*Mitf^mi-ew*/B6-*Mitf^mi-sp#* compound heterozygotes. (C) B6-*Mitf^Mi-Wh*/B6-*Mitf^mi-sp* and B6-*Mitf^Mi-Wh*/B6-*Mitf^mi-sp#* compound heterozygotes. (D) B6-*Mitf^mi-sp*/B6-*Mitf^mi* and B6-*Mitf^mi-sp#*/B6-*Mitf^mi* compound heterozygotes. Notice the dramatic suppression of the phenotype from near-white to black coat color. (E) B6-Mitf^mi-sp/B6-Mitf^mi-vga9. (F) B6-Mitf^mi-sp#/B6-Mitf^mi-vga9. (G) B6-Mitf^mi-sp/B6-Mitf^mi-rw and B6-Mitf^mi-sp#/B6-Mitf^mi-rw animals. (H) B6-Mitf^mi-sp/Mitf^mi-sp. (I) B6-Mitf^mi-sp #/Mitf^mi-sp#. (J) Graphical depiction of the *Mitf^mi-sl* mutation. (K) Antibody staining of melanocytes and eye sections from *Mitf^mi-sp* and *Mitf^mi-sl* tissues. The antibodies are 6A5, which recognizes the C-end of MITF, and a polyclonal rabbit anti-MITF antibody. Black scale bar for melanocytes: 20 μm and white scale bar for eye sections 500 μm. (L) DNA binding curves of recombinantly expressed human MITF-WT, MITF-sp, and MITF-sl proteins to M-box probe measured by fluorescence anisotropy at 165 mM KCl (blue line) and 300 mM KCl (yellow line). Each data point in the binding isotherms corresponds to an average of >5000 molecules. MITF-WT protein in circles, MITF-sp in square boxes, and MITF-sl in triangles. Error bars represent two standard deviations of fit error at each point. Source data are available online for this figure.

downstream residues (S69 in the case of S73 and S397, S401 and S405 in the case of S409) (Ngeow et al, 2018; Ploper et al, 2015). MITF has also been shown to be SUMOylated at K182 and K316 (Miller et al, 2005; Murakami and Arnheiter, 2005) and potentially ubiquitinylated at K201 and K265 (Shen et al, 2022; Xu et al, 2000). However, the biological function of the different post-translational modifications (PTMs) is largely unknown.

Individuals carrying the E318K germline mutation in MITF are predisposed to melanoma (Bertolotto et al, 2011; Yokoyama et al, 2011). The E318K mutation abolishes SUMOylation of the MITF protein at K316 (Bertolotto et al, 2011; Bonet et al, 2017; Yokoyama et al, 2011), and ChIP-seq studies have shown that the MITF-E318K protein has increased occupancy at known MITF-target genes compared to the wild-type protein but also binds to an increased number of genes. However, transcriptomic studies did not reveal major changes in gene expression (Bertolotto et al, 2011; Yokoyama et al, 2011). Mice carrying the E318K mutation exhibited slightly reduced pigmentation in both homo- and heterozygous conditions, whereas *Mitf^E318K*/+; *Braf^V600E*/+ mice had an increased number of nevi. Currently, it is not understood how the E318K mutation affects protein function or how it predisposes to melanoma.

Due to its obvious effects on pigmentation, MITF provides an excellent sensitized system for searching for suppressor mutations. In mice, over 40 different mutant alleles have been found in *Mitf* that can be arranged in an allelic series according to the severity of their phenotypic effects, as evidenced by coat color changes (Steingrímsson et al, 2004). At one end of the spectrum is the original and most severe allele *Mitf^mi* (Table 1; deletion of one of four arginines in the DNA-binding domain), which leads to a white coat, severe microphthalmia, and osteopetrosis and results in death at 3–4 weeks of age. At the other end of the spectrum is the mildest *Mitf* mutation, *Mitf^mi-spotted* (*Mitf^mi-sp*), which has no visible phenotype even when homozygous. The *Mitf^mi-sp* mutation lacks the alternative 18 bp exon 6A that encodes six amino acids upstream of the DNA-binding domain (Fig. 1A). Interestingly, the *Mitf^mi-sp* allele induces a white spotting phenotype when combined with other mutations at the locus (Arnheiter, 2010; Steingrímsson et al, 2004). For example, when the *Mitf^mi-sp* allele is mated to the original *Mitf^mi* mutation, the offspring exhibit a white coat with occasional grey areas and no microphthalmia. The intermediate coat pigmentation alterations obtained in compound heterozygotes with *Mitf^mi-sp* made this allele ideal for an N-ethyl-N-nitrosourea (ENU) mutagenesis screen for dominant suppressors or enhancers of the *Mitf* phenotype. Using this approach, we expected to find mutations in novel genes participating in the molecular pathways

through which *Mitf* regulates pigment cell development and melanogenesis. We isolated a mutation that suppressed the *Mitf* phenotype, but intriguingly, it is a derivative of the *Mitf^mi-sp* allele that lacks 104 residues of the carboxyl end (C-end). The induced *Mitf* suppressor mutation highlights the critical role of the IDR at the C-end of MITF in determining its stability, subcellular location, and transcriptional activity.

## Results

### Generation and analysis of an *Mitf* suppressor mutation

To screen for dominant mutations that suppress the *Mitf* phenotype, we crossed NAW-*Mitf^mi-ew*/*Mitf^mi-ew* females with C57BL/6J-*Mitf^mi-sp*/*Mitf^mi-sp* males that had been treated previously with the mutagen N-ethyl-N-nitrosourea (ENU). We screened for coat pigmentation changes in the F1 offspring (Fig. EV1A). While *Mitf^mi-sp* homozygotes have no visible coat color phenotype, animals homozygous for the *Mitf^mi-ew* mutation are white, severely microphthalmic, and exhibit mild hyperostosis (Steingrímsson et al, 2002); (Table 1). Compound heterozygotes for these two mutations have a "salt-and-pepper" body color with a white head, belly, and feet (Fig. 1B, left). ENU-treated F1 *Mitf^mi-sp*/*Mitf^mi-ew* heterozygotes were screened for coat pigmentation changes (Fig. EV1A). Of 63 NAW-*Mitf^mi-ew*/*Mitf^mi-ew* females, less than 50% produced progeny, resulting in a total of 470 offspring. In one of the matings, a deviant offspring female, marked by a '#', showed a considerably darker coat (near-black coat with pale ears, tails, and toes) compared to its littermates, suggesting the presence of a suppressor mutation (Fig. 1B, right).

When this *Mitf^mi-ew*/*Mitf^mi-sp#* female was bred to a C57BL/6J-*Mitf^mi-ew*/*Mitf^mi-ew* male, two classes of offspring resulted: white microphthalmic mice of the genotype *Mitf^mi-ew*/*Mitf^mi-ew* and mice of the genotype *Mitf^mi-ew*/*Mitf^mi-sp#* with the darkly pigmented phenotype of their mother (Fig. EV1B, left). This confirmed that the '#' mutation altering the *Mitf^mi-ew*/*Mitf^mi-sp* phenotype is dominant, at least for the combination of these two alleles. Also, because the above crosses did not yield mice with the phenotype expected for *Mitf^mi-ew*/*Mitf^mi-sp* mice, the novel suppressor mutation is likely closely genetically linked with *Mitf^mi-sp* or lies within the gene itself rather than on a different chromosome. Crossing the near-black *Mitf^mi-ew*/*Mitf^mi-sp#* mice to C57BL/6J animals only resulted in black offspring.

When the near-black *Mitf^mi-ew*/*Mitf^mi-sp#* mice were mated to white microphthalmic *Mitf^Mi-Wh*/*Mitf^Mi-Wh* homozygotes (Table 1), there were

**Table 1. The Mitf mutants used in this study.**

| Allele | Symbol | Mode of induction | Phenotype[a] Heterozygote | Homozygote | Lesion |
|---|---|---|---|---|---|
| Microphthalmia | $Mitf^{mi}$ | X-irradiation | Iris pigment less than in wild-type; occasional spots on belly, head, or tail | White coat, eyes small and red; deficiency of mast cells, incisors fail to erupt, osteopetrosis; inner ear defects | 3 bp deletion in basic domain |
| Spotted | $Mitf^{mi-sp}$ | Spontaneous | Normal (reduced tyrosinase activity in skin) | Normal (reduced tyrosinase activity in skin). MitfMi-Wh/Mitfmi-sp animals are light yellow with white spots on coat; eyes are pigmented | Additional cytosine in polypyrimidine tract; 18 bp exon missing |
| Eyeless-white | $Mitf^{mi-ew}$ | Spontaneous | Normal | White coat, eyes almost absent, eyelids never open | 25 amino acid deletion (splicing) |
| Spotless | $Mitf^{mi-sl}$ | ENU | Normal | "Brownish" coat color. Compound heterozygotes with other Mitf mutations show a more normal coat color than is seen with Mitfmi-sp mice. | Additional cytosine in polypyrimidine tract; 18 bp exon missing. In addition, Lys316STOP |
| White | $Mitf^{Mi-Wh}$ | spontaneous or X-irradiation | Coat color lighter than dilute (d/d); eyes dark ruby; spots on feet, tail and belly | White coat; eyes small and slightly pigmented; inner ear defects | I212N |
| Oak ridge | $Mitf^{Mi-or}$ | Gamma-irradiation | Pale ears and tail; belly streak or head spot | White coat, eyes small and red; incisors fail to erupt, osteopetrosis | R216K |
| Brownish | $Mitf^{Mi-b}$ | Spontaneous | Fur diluted brownish with pale ears and tail | White coat, reduced eye pigment, eyes of normal size | G244E |
| VGA-9 | $Mitf^{mi-vga9}$ | Transgene insertion | Normal | White coat, eyes red and small; inner ear defects | transgene insertion and 882 bp deletion |
| Red-eyed | $Mitf^{mi-rw}$ | White spontaneous | Normal | White with pigmented spot on head and rump; eyes small and red | Upstream genomic deletion |

[a]The phenotypes of the mutant alleles have been described by different researchers and to different extents. Features described for all the mutants are coat and eye color, eye size, and tooth and bone defects.

again two classes of offspring: the expected white mice with average eye size ($Mitf^{mi-ew}/Mitf^{Mi-Wh}$ heterozygotes) and "steel"-colored mice with pale ears, tail, toes, and a belly spot ($Mitf^{Mi-Wh}/Mitf^{mi-sp\#}$ animals, Fig. 1C, right). The coat color of the latter animals was darker than that of the corresponding $Mitf^{Mi-Wh}/Mitf^{mi-sp}$ animals (Fig. 1C, left). The color was even darker than $Mitf^{Mi-Wh}/Mitf$-WT mice, suggesting that the new mutant represents a gain-of-function compared to the wild-type. Similar effects were also seen when animals carrying the new $Mitf^{mi-sp\#}$ mutation were crossed to the dominant-negative $Mitf^{mi}$ (Fig. 1D), $Mitf^{Mi-or}$, and $Mitf^{Mi-b}$ mutations (Fig. EV1C–E) or the null mutation $Mitf^{mi-vga9}$ (Fig. 1E,F; Table 1). These observations showed that the suppressing effects of the new mutation were not restricted to the $Mitf^{mi-ew}$ allele and did not depend on the genetic background of the alleles tested (compare Fig. 1B to EV1B and EV1C to EV1D). However, the new mutation did not affect the coat color of the recessive $Mitf^{mi-rw}$ allele when compared to $Mitf^{mi-rw}/Mitf^{mi-sp}$ animals (Fig. 1G), reflecting the fact that the latter animals are already black and no further improvement in coat color is possible. No obvious changes were observed in eye size or bone development in any of the combinations since both phenotypes are normal in $Mitf^{mi-sp}$ homozygotes or their compound heterozygotes.

Intercrosses of $Mitf$-WT/$Mitf^{mi-sp\#}$ heterozygotes produced two classes of offspring: normal non-agouti (black) mice and mice with a diluted "brownish" coat color in a 3 to 1 ratio (compare Fig. 1H and I). Genotyping showed that the "brownish" animals were homozygous for $Mitf^{mi-sp\#}$. Thus, intriguingly, the new mutation results in a partial loss-of-function in homozygous condition, altering the coat color from black to brown.

## Molecular analysis of the $Mitf^{mi-sp\#}$ mutation

As the new mutation is either tightly linked to $Mitf$ on chromosome 6 or an intragenic mutation, we performed RT-qPCR and sequencing studies of $Mitf$ in total RNA isolated from homozygous $Mitf^{mi-sp\#}$ heart and kidney. This revealed the previously character-ized $Mitf^{mi-sp}$ mutation (the lack of the 18 bp alternative exon) (Steingrímsson et al, 1994). In addition, an A to T transversion was detected at nucleotide 1075 of the cDNA of the MITF-M isoform (Hodgkinson et al, 1993), replacing the codon for K316 with a stop-codon, resulting in premature truncation of the protein in exon 9 (Fig. 1J). The mutation was confirmed by sequencing genomic DNA from several animals. The # mutation is, therefore, an intragenic re-mutation of the $Mitf^{mi-sp}$ allele, now termed $Mitf^{mi-spotless}$ ($Mitf^{mi-sl}$), that leads to a protein, MITF$^{mi-sl}$, that lacks 104 residues of the C-end, including the K316 SUMOylation site (Miller et al, 2005; Murakami and Arnheiter, 2005), a caspase cleavage site (D345) (Larribere et al, 2005), phosphorylation sites implicated in the mTOR, GSK3β, and MAP kinase signal transduction pathways (S384, S397, S401, S405, and S409) (Vu et al, 2021) and the proposed transcription activation domain 3 (AD3) (Takeda et al, 2000; Fig. 1A).

To confirm that the C-end of MITF is missing from the $Mitf^{mi-sl}$ mutant, we stained primary melanocyte cultures generated from homozygous $Mitf^{mi-sl}$ and $Mitf^{mi-sp}$ embryos and eye sections from $Mitf^{mi-sl}$ and $Mitf^{mi-sp}$ mutants with the monoclonal antibody 6A5, which reacts with the C-end of MITF (Bharti et al, 2008a) and should not stain cells or tissues from $Mitf^{mi-sl}$ animals. As shown in Fig. 1K, the antibody did not give a signal in $Mitf^{mi-sl}$ melanocytes or eye sections, whereas $Mitf^{mi-sp}$ melanocytes and eye sections

exhibited clear nuclear staining. In contrast, eye sections from both genotypes stained positive with a polyclonal rabbit anti-MITF antibody. This shows that the carboxyl-end (C-end) of MITF is absent from melanocytes and eyes of *Mitf^{mi-sl}* homozygotes.

## The MITF-sl protein forms stable dimers and binds DNA

We determined the dimerization and DNA-binding ability of the MITF^{mi-sp} and MITF^{mi-sl} proteins. In the discussion below, we simplify the nomenclature of the mutants to MITF-sp, MITF-sl and so on. All MITF constructs used in this project were generated in the mouse MITF-M cDNA, except the constructs used for direct DNA binding and structural studies where the human MITF-M cDNA was used (see later); all residues mutated are conserved between the two species. We co-expressed Flag-tagged versions of the non-DNA binding mutant proteins MITF-Wh, MITF-mi, and MITF-ew (Table 1) together with the MITF-WT-GFP, MITF-sp-GFP, or MITF-sl-GFP proteins in A375P melanoma cells which express little endogenous MITF (Wouters et al, 2020) followed by co-immunoprecipitation (co-IP) using FLAG-antibodies. The results showed that the non-DNA binding MITF mutant proteins successfully immunoprecipitated all three GFP-labeled proteins (Appendix Fig. S1A). We further confirmed the interactions between MITF-mi-Flag and GFP-tagged MITF-WT, -sp, and -sl proteins by Blue native PAGE (Wittig et al, 2006). The results suggest that the MITF-mi protein as well as the MITF-WT, -sp, and -sl proteins can form both hetero- and homodimers (Appendix Fig. S1B,C).

We measured the DNA binding affinity of recombinant human MITF-WT, -sp, and -sl proteins to a fluorescently labeled M-box probe by measuring changes in anisotropy on individual molecular complexes with single-molecule spectroscopy. Quantification gave dissociation constants for all constructs at 300 mM KCl that were within one standard deviation from each other (Fig. 1L; Table 2), while the affinities at 165 mM KCl ($K_D < 250$ pM) were too high to compare accurately. We also used single-molecule Förster resonance energy transfer (smFRET) and fluorescence correlation spectroscopy (FCS) as two additional and independent measures of MITF interactions with DNA (Fig. EV2A–D). All three methods yielded similar dissociation constants for all constructs at 300 mM KCl (Figs. 1L and EV2A–D; Table 2), and similar to that reported for the DNA binding domain alone (Möller et al, 2019). From the FCS data, we observed a smaller change in diffusion time upon DNA binding of MITF-sl than -sp and -WT, consistent with its smaller size (Fig. EV2A; Table 2). The electrophoretic mobility shift assay (EMSA) also showed similar steady-state affinity binding to M-box DNA of mouse MITF-WT, -sp, and -sl (Fig. EV2E). Co-translating the non-DNA binding MITF-mi with MITF-WT, MITF-sp, and MITF-sl, thus allowing heterodimerization before EMSA showed that increasing amount of the MITF-mi protein

interfered with the DNA binding of all three proteins. However, MITF-mi was more effective at interfering with MITF-WT than with either of the mutant proteins lacking exon 6A (Fig. EV2E), which is consistent with previous observations (Pogenberg et al, 2012). A slightly different picture emerged when the proteins were translated separately and subsequently mixed together before the EMSA. Again, the MITF-mi protein was more effective at interfering with the DNA binding of the MITF-WT protein than with that of the MITF-sp and MITF-sl proteins. However, it was even less effective at interfering with the DNA-binding of the MITF-sl protein (Fig. EV2F) than MITF-sp. This suggests that the MITF-sl homodimers are more stable than either the MITF-WT or MITF-sp homodimers and, thus, less prone to interference by a dominant-negative protein such as the MITF-mi protein.

## The MITF-sl protein affects gene expression

To investigate the effects of the *Mitf^{mi-sl}* mutation on gene expression, we induced the expression of mouse MITF proteins at an equal level in A375P cells and harvested RNA at regular intervals for qPCR analysis. The fold change of MITF target genes in cells overexpressing either MITF-WT or MITF-sl was compared to those expressing EV-FLAG-HA followed by normalization to the proportion of MITF protein retained in the nucleus. Consistent with previous work (Ballesteros-Álvarez et al, 2020; Louphrasitthi-phol et al, 2020), the expression of the endogenous human *MITF* mRNA was considerably reduced over the 36 h sampling period upon overexpressing mouse MITF-WT and MITF-sl proteins (Fig. EV3A). The expression of the *CDH2* and *NRP1* genes, both of which have been shown to be repressed by MITF (Dilshat et al, 2021) was also significantly reduced upon overexpression of MITF-WT and MITF-sl (Fig. EV3B,C). While MITF-WT activated the expression of *PMEL* and *TRIM63*, MITF-sl exhibited about half the activating ability of WT (Fig. EV3D,E). Interestingly, the MITF-sl protein was severely impaired in activating the expression of the pigmentation genes *TYRP1, MLANA, TYR, and DCT* (Fig. EV3F–I). We also performed CUT&RUN experiments to determine how MITF-sl affects the genome-wide occupancy of MITF. MITF-sl had a higher number of peaks than MITF-WT, with 10,636 peaks ($P < 0.01$) exhibiting statistically significant differences ($P < 0.01$) between MITF-WT and MITF-sl (Fig. EV3J,K; Dataset EV1). Gene ontology analysis indicates that the peaks significantly different between MITF-WT and MITF-sl are associated with genes involved in axonogenesis, axon development, cell growth, and positive regulation of MAPK cascade biological pathways (Fig. EV3L). Importantly, MITF-sl has altered binding to genes which were changed in expression compared to MITF-WT (Fig. EV3A–I), including NRP1, CDH2, PMEL, TRIM63, TYRP1, MLANA, and DCT. Our results suggest that the 316–419 domain is critical for selective genome occupancy and transcriptional activation of MITF.

## The *Mitf^{mi-sl}* mutation affects protein stability and localization

The effects of the *Mitf^{mi-sl}* mutation on protein stability were investigated by expressing Flag-tagged (at C-end) MITF-WT, MITF-sp, and MITF-sl proteins in a doxycycline (dox)-inducible vector transfected into A375P melanoma cells. Expression of the

**Table 2. DNA binding affinity of recombinant human MITF-WT, MITF-sp, and MITF-sl to M-box.**

| Construct | Anisotropy | FCS | FRET |
|-----------|------------|-----|------|
| hMITF-WT | 1.9 ± 0.6 nM | 4.5 ± 1.4 nM | 1.6 ± 1.4 nM |
| hMITF-sp | 1 ± 0.4 nM | 5.8 ± 3.5 nM | 1.7 ± 0.9 nM |
| hMITF-sl | 0.9 ± 0.4 nM | 3.1 ± 1.6 nM | 1.6 ± 0.5 nM |

proteins was equalized by treating the cells with varying concentrations of dox for 24 h. The cells were then treated with the translation inhibitor cycloheximide (CHX) for different periods and harvested to visualize MITF protein by Western blotting. The MITF protein is observed as two bands where the upper band is phosphorylated at S73 (hereafter referred to as pS73-MITF), and the lower band is not phosphorylated at S73 (hereafter referred to as S73-MITF) (Fock et al, 2019; Ngeow et al, 2018). The bands on the Western blot were quantitated and the changes in protein concentration were plotted over time. This data was used to calculate protein half-life, defined as the time required to reduce the initial protein abundance to 50%. The MITF-WT and MITF-sp proteins had comparable half-lives of 3.2 h for pS73-MITF and 1.2 h for S73-MITF (Fig. 2A,B). Critically, the MITF-sl protein was considerably less stable, with half-lives of 1.2 and 0.4 h for the pS73 and S73 forms, respectively (Fig. 2A,B). To confirm that exon 6A does not contribute substantially to MITF stability, we measured the stability of the pS73 and S73 versions of MITF-Wh with and without exon 6A and found that they were not significantly different (Appendix Fig. S2A,B). When overexpressed in the 501Mel and SKmel28 melanoma cell lines (which express high levels of MITF), the MITF-sl protein was also less stable than the MITF-WT and MITF-sp proteins, regardless of the S73 phosphorylation status (Appendix Fig. S2C,D). We also tested the stability of proteins carrying the Flag tag at the N-end or GFP tag at the C-end. In all cases, MITF-sl protein was less stable than MITF-WT and MITF-sp, regardless of the fusion tags and their location (Appendix Fig. S2E–G). Our finding, therefore, suggests that the absence of the 316–419 domain significantly reduces the stability of MITF. Furthermore, in all cases, the S73 MITF (lower band) protein was degraded faster than the pS73 form (upper band). Alternatively, the S73 protein may be phosphorylated and thus become pS73 during the experiment.

To determine the effect of MITF-sl on subcellular localization, we used dox-inducible A375P melanoma cells overexpressing MITF-Flag fusion proteins and performed cellular fractionation. After inducing expression of MITF for 24 h, the nuclear and cytoplasmic fractions were separated as described (Ramsby and Makowski, 1999; Senichkin et al, 2021), and MITF proteins were characterized by Western blotting. For the MITF-WT and MITF-sp proteins, both pS73 and S73 bands were observed at similar ratios in the nuclear and cytoplasmic fractions (Fig. 2C,D). However, for MITF-sl, both pS73 and S73 bands were predominantly located in the nucleus (Fig. 2C,D). The same results were observed when the MITF-WT, MITF-sp, and MITF-sl proteins were overexpressed in the 501Mel and SKmel28 melanoma cell lines (Appendix Fig. S3A,B). Flag-tagging the MITF protein at the N-end or replacing the C-end Flag with GFP also resulted in a significantly increased nuclear presence of the MITF-sl protein (Appendix Fig. S3C,D). Co-IP showed that nuclear accumulation of MITF-sl protein was not due to effects on interactions with 14-3-3 protein (Appendix Fig. S3E), which has been shown to interact with MITF phosphorylated at S173 and lead to the retention of MITF in the cytosol in osteoclasts (Bronisz et al, 2006). To determine if the six amino acids encoded by exon 6A were able to mediate nuclear localization, MITF lacking (−) or containing (+) this exon was transiently expressed in A375P cells. No difference was observed in the distribution of MITF between the nuclear and cytoplasmic fractions of the MITF-Wh and MITF-Wh(−) or MITF-sl(+) and

MITF-sl constructs (Appendix Fig. S3F). Taken together, we conclude that residues 316–419 of MITF, but not exon 6A or the tags, affect MITF subcellular localization.

## MITF^mi-sl translocates its Mitf partners into the nucleus and improves its own stability

To determine if the MITF-sl protein might affect the subcellular localization of the non-DNA binding mutant MITF proteins MITF-mi, MITF-ew and MITF-Wh, they were transiently co-overexpressed with the MITF-sl protein followed by nuclear fractionation. As before, a significant portion of the pS73- and S73-MITF-sl proteins was observed in the nucleus (Fig. 2E–G). Consistent with previous reports (Fock et al, 2019; Takebayashi et al, 1996), stably expressed MITF-mi and MITF-ew mutant proteins are primarily present in the cytoplasm (Appendix Fig. S4A, compare with Fig. 2C for MITF-WT or MITF-Wh proteins). In contrast to stably expressed proteins (Appendix Fig. S4A), transiently expressed MITF-mi or MITF-ew showed equal distribution between cytoplasm and nucleus; no significant difference was noted between stable and transient expression of the MITF-Wh protein (Fig. 2E,F). However, when co-expressed with the MITF-sl protein, they were all significantly translocated into the nucleus (Fig. 2E,G). Similar results were observed in cells co-expressing MITF-sl and MITF-WT (Appendix Fig. S4B–D). Our data, therefore, strongly suggest that the MITF-sl protein can dimerize with both mutant and WT proteins and induce nuclear localization of its partner by either translocating the dimer to the nucleus or keeping it from leaving the nucleus.

We assessed the stability of the MITF-sl protein in cells also expressing either MITF-Wh, MITF-mi, or MITF-ew. Interestingly, the stability of both pS73 and S73 MITF-sl was considerably increased in the presence of the MITF-Wh, MITF-mi, and MITF-ew proteins, with the most pronounced effect observed in cells also expressing MITF-mi and MITF-ew (around 2.5-fold increase for pS73 and 3.5-fold increase for S73) (Fig. 2H,I). However, the stability of the MITF-Wh, MITF-mi, and MITF-ew dimeric partner proteins themselves remained unchanged upon co-expression of MITF-sl as compared to the condition when they were expressed in the absence of MITF-sl (Fig. 2H,I). The stability of pS73 and S73 versions of MITF-sl was also significantly improved when co-expressed with MITF-WT (Appendix Fig. S4E,F). To eliminate the possibility that we saturated the degradation machinery in the cells, we co-transfected the cells with MITF-sl-GFP and MITF-sl-Flag and measured the stability of MITF-sl-Flag protein. The results showed that the stability of MITF-sl-Flag was not affected by the presence of MITF-sl-GFP (Appendix Fig. S4G). Taken together, our data suggest that the MITF-sl protein forms dimers with the MITF-Wh, MITF-mi, and MITF-ew proteins, which then drags them into the nucleus or prevents them from leaving the nucleus, leading to increased stability of the MITF-sl protein itself (without, however, changing the stability of the partner proteins). On balance, this may increase the formation of DNA-binding MITF-sl homodimers after dissociation from their dimeric partners, and so explain the genetic suppression effect observed in vivo. The effects on protein stability and gene expression changes induced in the presence of MITF-sl alone might explain the hypomorphic effect in *Mitf^mi-sl* homozygotes. That this hypomorphic effect is not seen in compound heterozygotes with the non-DNA binding mutants may be due to the balance between nuclear import/export, effects on stability, and

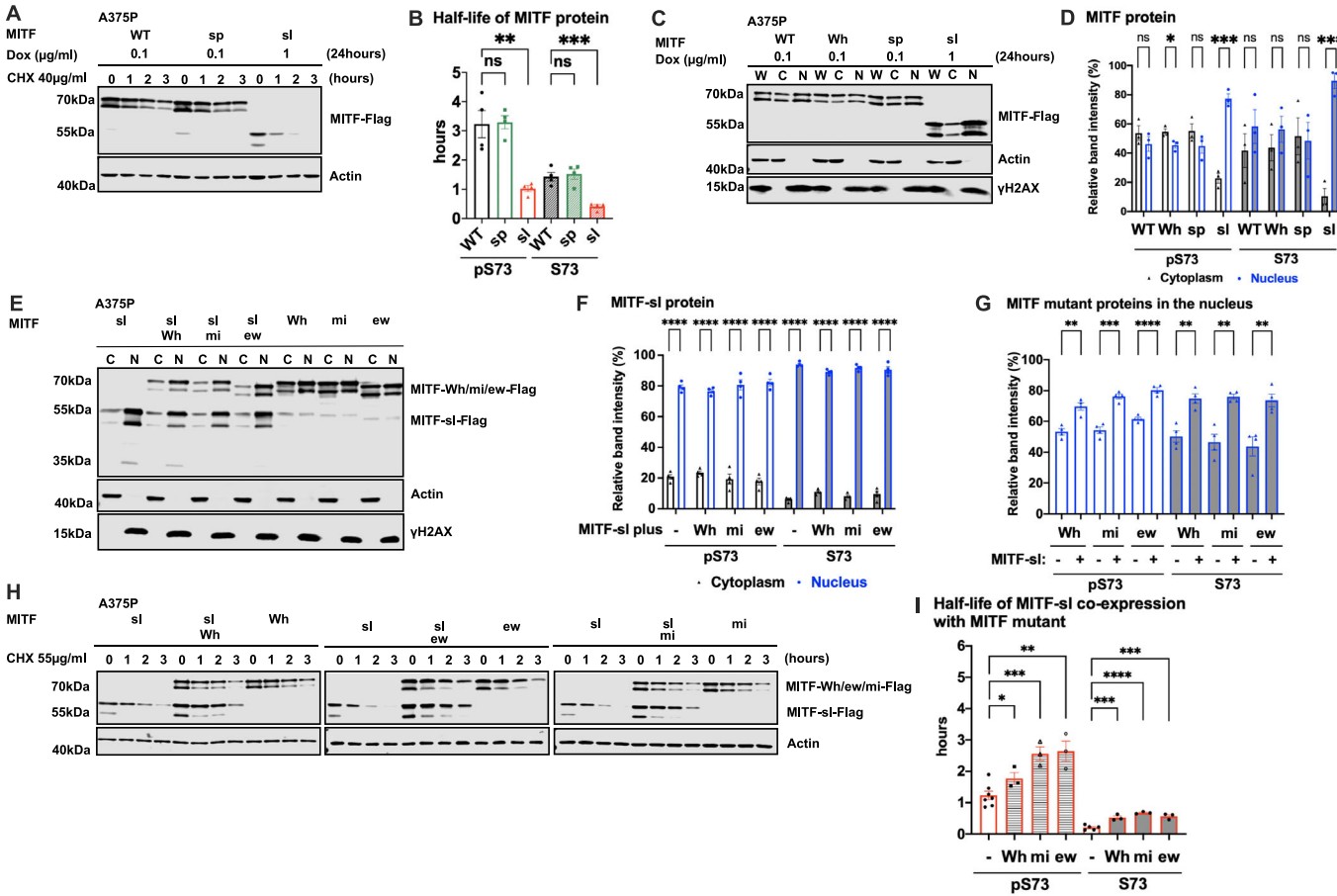

**Figure 2. The carboxyl-domain of Mitf controls RNA and protein levels as well as its subcellular localization.**

(**A**) Western blot analysis of the Mitf-Flag proteins upon cycloheximide treatment. The dox-inducible A375P cells expressing the MITF-WT, MITF-sp, and MITF-sl proteins were treated with doxycycline for 24 h to induce similar expression of the indicated mutant MITF proteins before treating them with 40 µg/ml cycloheximide (CHX) for 0, 1, 2, and 3 h. The blots were stained using Flag antibody and protein quantitated using the Odyssey imager and ImageJ. Actin was used as a loading control. (**B**) Half-life analysis of the indicated pS73- and S73-MITF proteins over time after CHX treatment in A375P melanoma cells. The relative MITF protein levels to T0 were calculated, and non-linear regression analysis was performed. Error bars represent SEM of at least three independent experiments. Statistically significant differences were calculated using unpaired Student's t-test. P values for the pS73-MITF form of WT compared to sp and sl were 0.9077 and 0.0037, respectively. P values for the S73-MITF form of WT compared to sp and sl were 0.7085 and 0.0007, respectively. (**C**) Western blot analysis of subcellular fractions isolated from A375P melanoma cells induced for 24 h to overexpress different MITF mutant proteins. MITF-WT, MITF-Wh, MITF-sp, and MITF-sl protein in whole cell lysate (W), cytoplasmic (C), and nuclear (N) fractions were visualized using FLAG antibody. Actin and γH2AX were loading controls for cytoplasmic and nuclear fractions, respectively. (**D**) Intensities of the indicated pS73- and S73-MITF protein bands in the cytoplasmic and nuclear fraction from the western blot analysis in (**C**) were quantified separately with ImageJ software and are depicted as percentages of the total amount of protein present in the two fractions. Error bars represent SEM of three independent experiments. Statistically significant differences were calculated using unpaired Student's t test. P values for the pS73-MITF form of WT, Wh, sp, and sl were 0.3512, 0.040, 0.2150, and 0.0003, respectively. P values for the S73-MITF form of WT, Wh, sp, and sl were 0.3733, 0.3761, 0.8689, and 0.0004, respectively. (**E**) Western blot analysis of subcellular fractions isolated from A375P cells transiently co-overexpressing the MITF-sl protein with the MITF-Wh, MITF-mi, and MITF-ew mutant MITF proteins. MITF proteins in cytoplasmic (C) and nuclear (N) fractions were visualized using FLAG antibody. Actin and γH2AX were loading controls for cytoplasmic and nuclear fractions, respectively. The MITF-sl protein migrates as a doublet at 50–55 kDa, whereas the other mutants migrate at 65–70 kDa. (**F**) The intensities of the pS73- and S73- MITF-sl protein in the cytoplasmic and nuclear fractions from western blot analysis (**E**) were quantified separately with ImageJ software and are depicted as percentages of the total protein present in the two fractions. Error bars represent SEM of at least three independent experiments. Statistically significant differences were calculated using unpaired Student's t test. P values for the pS73- and S73-MITF-sl co-expression with empty vector (-) Wh, mi, and ew were ****P < 0.0001. (**G**) Quantification of band intensities of the pS73- and S73-versions of the MITF-Wh, MITF-mi, and MITF-ew proteins as determined from western blots (**E**) in the nuclear fractions of A375P cells transiently co-overexpressing the MITF-sl protein with the indicated MITF mutant proteins. The intensities are depicted as percentages of the total amount of protein present in the two fractions. Error bars represent SEM of at least three independent experiments. Statistically significant differences were calculated using unpaired Student's t test. P values for the pS73-MITF form of Wh, mi, and ew with or without co-expressing with MITF-sl were 0.0021, 0.0002, and <0.0001, respectively. P values for the S73-MITF form of Wh, mi, and ew with or without co-expressing with MITF-sl were 0.0023, 0.0015, and 0.0072, respectively. (**H**) Western blot analysis of the degradation of the MITF-sl protein in the presence of non-DNA binding MITF mutations (MITF-Wh, MITF-mi, and MITF-ew). The A375P cells were transiently co-transfected with MITF-sl and either MITF-mi, MITF-ew, or MITF-Wh for 24 h before being treated with 55 µg/ml CHX. The amount of MITF protein was then compared by western blot using FLAG antibody. Actin was used as a loading control and normalized to the expression of MITF protein expression. The band intensities were quantified using ImageJ software. (**I**) Half-life analysis of the indicated pS73- and S73-MITF proteins over time after CHX treatment. The relative MITF protein levels to T0 were calculated, and non-linear regression analysis was performed. Error bars represent SEM of at least three independent experiments. Statistically significant differences were calculated using unpaired Student's t test. P values for the pS73-MITF-sl form with or without co-expressing with Wh, mi, and ew were 0.0185, 0.0005, and 0.0015, respectively. P values for the S73-MITF-sl form with or without co-expressing with Wh, mi, and ew were 0.0005, <0.0001, and 0.0009, respectively. Source data are available online for this figure.

the rate of dissociation of MITF-sl from its dimeric partner and subsequent effects on transcription.

## Effects on nuclear localization and stability are encoded in the carboxyl-domain

To determine which regions within the C-end of MITF contain its nuclear retention properties, we generated truncated versions of MITF-sp with Flag-tag fusion at the C-end in our inducible vector system (schematic diagram in Fig. 3A). The S73-MITF-WT and S73-MITF-sp-Δ326–377 proteins are distributed equally between the cytoplasmic and nuclear fractions; the pS73-MITF-sp-Δ326–377 was slightly more cytoplasmic (Fig. 3B,C). In contrast, a significant portion of the MITF-sp-326* and MITF-sp-378* proteins was present in the nuclear fraction (Fig. 3B,C), suggesting that the 378–419 domain, including the phosphorylation sites indicated in Fig. 3A, plays an essential role in controlling the nuclear localization of MITF. Interestingly, the non-phosphorylatable alanine mutation at S409 led to slightly more nuclear localization of the pS73 MITF form, whereas the single S384A, S397A, S401A, and S405A mutations did not alter MITF nuclear localization (Appendix Fig. S5A,B). However, the quadruple S397/401/405/409A mutation in MITF-sp (MITF-sp-4A) or MITF-WT (MITF-WT-4A) led to increased nuclear localization of the respective proteins (Appendix Fig. S5A,B). This suggests that the phosphorylation cascade at the C-end may be involved in the cytoplasmic retention of MITF but that other elements within the C-end must also be involved.

The MITF-sl protein was more nuclear than the MITF-sp-326* and MITF-sp-378* proteins (Fig. 3B,C), suggesting that the 316–326 domain must also be involved in regulating nuclear localization. However, the MITF-sp-Δ316–326 construct, which lacks the K316 SUMO-site and adjacent residues, did not alter the cytoplasmic-nuclear distribution of MITF (Appendix Fig. S5C,D). This suggests that residues 326–419 contain a major signal for mediating nuclear export of MITF and that residues 316–326 also contribute. Intriguingly, the MITF-mi and MITF-ew proteins containing the 316*, 378*, or Δ316–326 mutations were more nuclear than their full-length counterparts (Appendix Fig. S6A,B).

Previous work has shown that treatment with 12-O-tetradeca-noylphorbol-13-acetate (TPA), an agent known to induce ERK kinase activity, leads to phosphorylation of S73 of MITF and shifts the protein to the cytoplasm (Ngeow et al, 2018). Consistent with that, TPA treatment promoted S73 phosphorylation (as seen by the almost exclusive presence of the upper MITF-band) of MITF-WT and shifted the protein out of the nucleus (Fig. 3D,E). The MITF-sl protein was also phosphorylated at S73 but, as before, it mostly stayed in the nucleus. The MITF-sp-Δ326–377, MITF-sp-Δ316–326, and MITF-sp-378* proteins were also phosphorylated at S73, but a large proportion of these proteins was located in the cytoplasm after TPA treatment. The MITF-sp-326* protein was equally distributed between the two compartments (Fig. 3D,E). These data suggest that MITF has a nuclear export signal in the C-end, which may act independently of the one involving S69 and S73. The LocNES algorithm (Xu et al, 2015) predicts a couple of nuclear export signals (NESs) in the C-end of MITF spanning residues 336–350 (NES1) and 374–388 (NES2). To test their role, we generated a fusion of MITF-sl to either NES1 or NES2 or both NES1 and NES2 (schematic diagram in Fig. EV4A) in our inducible vector system and performed cell fractionation. Fusions of NES1

and/or NES2 slightly increased the proportion of MITF-sl in the cytoplasm, regardless of S73 phosphorylation status (Fig. EV4B,C). Upon TPA treatment, pS73-MITF-sl-NES1, -NES2, and NES1-NES2 were significantly exported to the cytoplasm, as opposed to pS73-MITF-sl (Fig. EV4B,D). Our findings suggest that NES1 and NES2 are involved either in nuclear export of MITF or in blocking its import.

To determine which regions within the C-end of MITF are essential for mediating effects on stability, we performed protein stability assays using the MITF deletion constructs in the presence of CHX. The results showed that, again, the pS73 form of MITF-WT, MITF-sl, MITF-sp-326*, MITF-sp-Δ316–326, and MITF-sp-378* proteins was considerably more stable than the corresponding S73 proteins (Fig. 3F,G). It also showed that the MITF-sp-326*, MITF-sp-378*, and MITF-sp-Δ316–326 proteins were less stable than the MITF-WT protein. However, MITF-sl still showed the most rapid degradation upon CHX treatment of all proteins tested (Fig. 3F,G). The results suggest that the 316–326 and 378–419 domains are important for nuclear localization and MITF stability. The effects of the 378–419 domain on MITF localization are not due to a single phosphorylation site at the C-end since S384A, S397A, S401A, S405A, and S409A did not significantly affect the stability of MITF-sp, nor did their combination in the 4A mutant construct (Appendix Fig. S7).

To further investigate the role of the 316–419 domain in mediating MITF protein stability, we determined the stability of the non-DNA binding MITF-mi, MITF-mi-316*, MITF-ew, and MITF-ew-316* proteins. Although the pS73-MITF-mi and pS73-MITF-ew proteins were slightly more stable than pS73-MITF-WT, the stability of S73-MITF-mi and S73-MITF-ew did not significantly differ from MITF-WT (Appendix Fig. S8A,B). Meanwhile, the double mutant proteins (i.e., MITF-mi-316* and MITF-ew-316*) exhibited increased presence in the nucleus (Appendix Fig. S6) yet had similar stability as MITF-WT (Appendix Fig. S8C,D). This suggests that the ability to bind to DNA in concert with MITF C-end might be important for controlling MITF stability and triggering the degradation process.

## MITF is mainly degraded through a ubiquitin-mediated proteasome pathway in the nucleus

To determine which degradation pathway is responsible for MITF degradation, we treated the cells with the ubiquitin-proteasomal inhibitor MG132 and the lysosomal inhibitor Baf-A1 together with CHX for 3 h. Treatment with MG132 and CHX increased the stability of the MITF-WT, MITF-sp, and MITF-sl proteins, whereas Baf-A1 and CHX treatment did not (Fig. 4A,B). Treatment with MG132 or Baf-A1 without CHX showed a significant increase in the intensity of the pS73 band of the MITF-sp and MITF-sl proteins, though not MITF-WT protein (Fig. 4C,D). Interestingly, the S73 bands of MITF-WT, MITF-sp, and MITF-sl showed a considerable increase after MG132 treatment (approximately 2.4-, 2.7-, and 6.7-fold increase, respectively), but the increase was much less pronounced or even non-significant (in the case of S73-MITF-sl) upon Baf-A1 treatment (Fig. 4D). This suggests that the ubiquitin-proteasome pathway is the primary degradation machinery for MITF.

To determine where the proteasomal degradation pathway takes place, we treated MITF-WT expressing cells with both MG132 and TPA. As shown in Fig. 4E, TPA treatment significantly increased the total MITF protein compared to vehicle controls, suggesting that

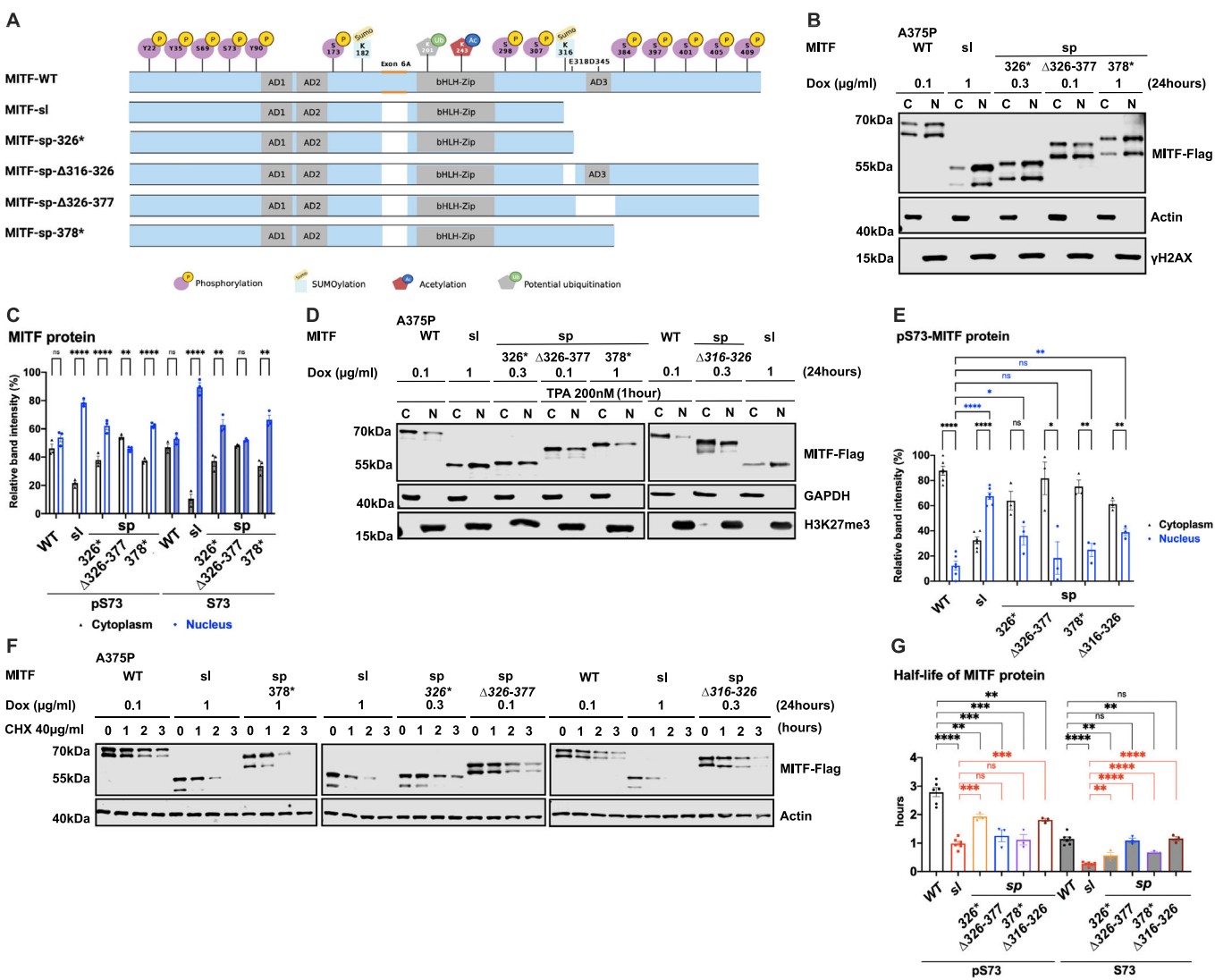

**Figure 3. The carboxyl-domains of Mitf control its nuclear localization and stability.**

(A) Schematic of MITF-sp truncation constructs. C-term truncations were generated by introducing stop codons at position Q326 or L378 or by deleting fragments 326–377 or 316–326. MITF-sp-326* introduces a stop-codon at residue 326 and, therefore, contains the SUMO-site at 316; MITF-sp-Δ326–377 lacks the tentative activation domain AD3; MITF-sp-Δ316–326 lacks the SUMO-site and adjacent amino acids; MITF-sp-378* lacks the series of phosphorylation sites at the carboxyl-end of the protein. (B) Western blot analysis of subcellular fractions isolated from A375P melanoma cells induced to overexpress the different MITF mutant proteins fused with Flag-tag at C terminus for 24 h. MITF-WT, MITF-sl, MITF-sp-326*, MITFmi-sp-378*, and MITF-sp-Δ326–377 in cytoplasmic (C) and nuclear (N) fractions were visualized using FLAG antibody. Actin and γH2AX were loading controls for cytoplasmic and nuclear fractions, respectively. (C) The intensities of the indicated pS73 MITF and S73 MITF proteins from the cytoplasmic and nuclear fractions of the western blot analysis in (B) were quantified separately with ImageJ software and are depicted as percentages of the total amount of protein present in the two fractions. Error bars represent SEM of three independent experiments. Statistically significant differences were calculated using unpaired Student's *t* test. *P* values for the pS73-MITF form of WT, sl, 326*, Δ326–377, and 378* were 0.1455, <0.0001, 0.0033, 0.0054, and <0.0001. *P* values for the S73-MITF form of WT, sl, 326*, Δ326–377, and 378* were 0.1302, <0.0001, 0.0072, 0.3957, and 0.0021, respectively. (D) Western blot analysis of subcellular fractions isolated from A375P melanoma cells induced for 24 h to overexpress the different MITF mutant proteins before treatment with TPA at 200 nM for 1 h. MITF-WT, MITF-sl, MITF-sp-326*, MITF-sp-Δ326–377, MITF-sp-Δ316–326, and MITF-sp-378* protein in cytoplasmic (C) and nuclear (N) fractions were visualized using FLAG antibody. Actin or GAPDH and γH2AX or H3K27me3 were loading controls for cytoplasmic and nuclear fractions, respectively. (E) Intensities of the indicated pS73-MITF proteins from the western blot analysis in (D) in the cytoplasmic and nuclear fractions from the cell treated with TPA were quantified separately with ImageJ software and are depicted as percentages of the total amount of protein present in the two fractions. Error bars represent SEM of three independent experiments. Statistically significant differences were calculated using unpaired Student's *t* test. *P* values for the pS73-MITF form of WT, sl, 326*, Δ326–377, 378*, and Δ316–326 were <0.0001, <0.0001, 0.0573, 0.0258, 0.0025, and 0.0050. *P* values for the S73-MITF form of sl, 326*, Δ326–377, 378*, and Δ316–326 compared to pS73-MITF-WT in the nuclear fraction were <0.0001, 0.0125, 0.5609, 0.0835, and 0.0019, respectively. (F) Western blot analysis of the MITF proteins from dox-induced A375P cells after treating them with 40 μg/ml CHX for 0, 1, 2, and 3 h. The MITF proteins were visualized by western blot using FLAG antibody. Actin was used as a loading control. The band intensities were quantified using ImageJ software. (G) Half-life analysis of the indicated pS73- and S73-MITF proteins over time after CHX treatment. The MITF protein levels relative to T0 were calculated, and non-linear regression analysis was performed. Error bars represent SEM of at least three independent experiments. Statistically significant differences were calculated using unpaired Student's *t* test. *P* values for the pS73-MITF form of sl, 326*, Δ326–377, 378*, and Δ316–326 compared with pS73-MITF-WT were <0.0001, 0.0093, 0.0008, 0.0004, and 0.0046, respectively. *P* values for the S73-MITF form of sl, 326*, Δ326–377, and Δ316–326 compared to S73-MITF-WT were <0.0001, 0.0042, 0.7240, 0.0064, and 0.8886, respectively. *P* values for the pS73-MITF form of 326*, Δ326–377, 378*, and Δ316–326 compared with pS73-MITF-sl were 0.0001, 0.1748, 0.4561, and 0.0002 respectively. *P* values for the S73-MITF form of 326*, Δ326–377, 378*, and Δ316–326 compared to S73-MITF-sl were 0.0077, <0.0001, <0.0001, <0.0001, respectively. Source data are available online for this figure.

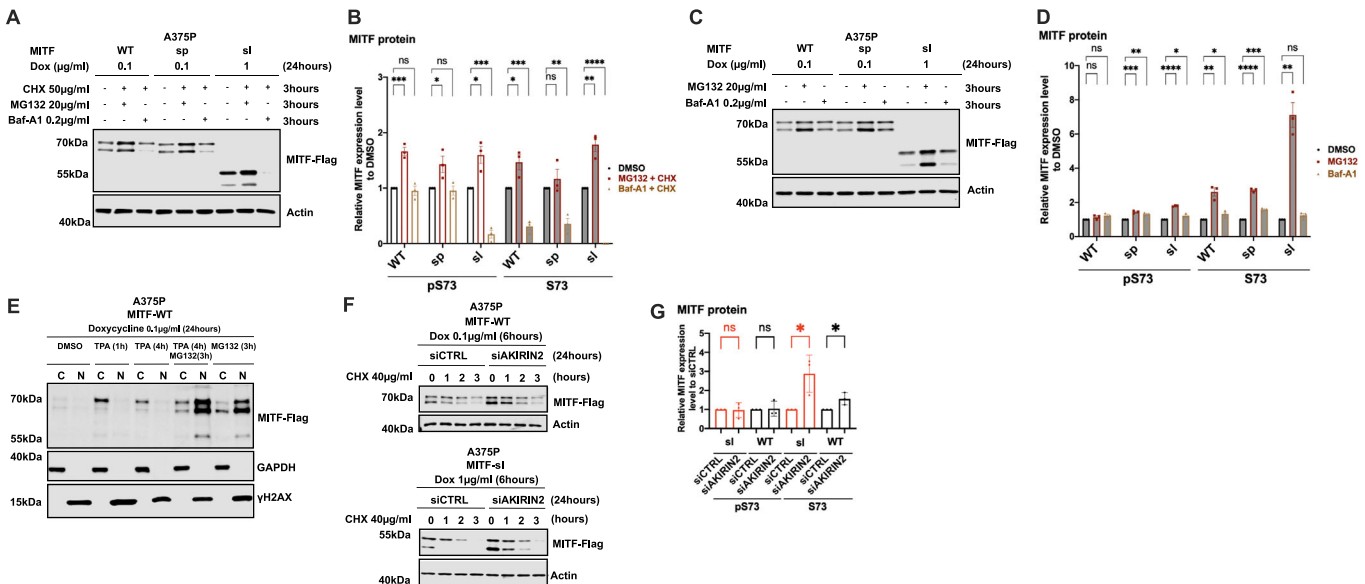

**Figure 4. MITF is mainly degraded through the proteasome pathway in the nucleus.**

(A) Western blot analysis of the MITF-WT, MITF-sp, and MITF-sl proteins. Expression was induced for 24 h in A375P cells treated with 50 µg/ml CHX in the presence of either DMSO or 20 µg/ml MG132 or 0.2 µg/ml Baf-A1 for 3 h. The MITF protein was then visualized by western blot using FLAG antibody. Actin was used as a loading control. The band intensities were quantified using ImageJ software. (B) The indicated pS73- and S73-MITF protein band intensities from western blot analysis (A) were quantified separately with ImageJ software and are depicted relative to DMSO. Error bars represent SEM of at least three independent experiments. Statistically significant differences were calculated using unpaired Student's t test. Compared between DMSO and MG132 treated conditions in the presence of CHX, p values for the pS73-MITF form of WT, sp, and sl were 0.0008, 0.0443, and 0.0176, respectively. P values for the S73-MITF form of WT, sp, and sl were 0.0279, 0.3753, and 0.0035, respectively. Compared between DMSO and Baf-A1 treated conditions in the presence of CHX, P values for the pS73-MITF form of WT, sp, and sl were 0.5988, 0.6219, and 0.0003, respectively. P values for the S73-MITF form of WT, sp, and sl were 0.0005, 0.0028, and <0.0001, respectively. (C) Western blot analysis of the MITF-WT, MITF-sp, and MITF-sl proteins. Expression was induced for 24 h in A375P cells treated with either DMSO or 20 µg/ml MG132 or 0.2 µg/ml Baf-A1 for 3 h. The MITF protein was then visualized by western blot using FLAG antibody. Actin was used as a loading control. The band intensities were quantified using ImageJ software. (D) The indicated pS73- and S73-MITF protein band intensities from western blot analysis (C) were quantified separately with ImageJ software and are depicted relative to DMSO. Error bars represent SEM of at least three independent experiments. Statistically significant differences were calculated using unpaired Student's t test. Compared between DMSO and MG132 treated conditions, P values for the pS73-MITF form of WT, sp, and sl were 0.1532, 0.0007, and <0.0001, respectively. P values for the S73-MITF form of WT, sp, and sl were 0.0026, <0.0001, and 0.0011, respectively. Compared between DMSO and Baf-A1 treated conditions in the presence of CHX, P values for the pS73-MITF form of WT, sp, and sl were 0.0558, 0.0043, and 0.0372, respectively. P values for the S73-MITF form of WT, sp, and sl were 0.0427, 0.0005, and 0.0948, respectively. (E) Western blot analysis of subcellular fractions isolated from A375P melanoma cells induced to overexpress MITF-WT protein before treating with either 200 nM TPA for 1 or 4 h or 40 µg/ml MG132 for 3 h or 200 nM TPA for 1 h and then adding 40 µg/ml MG132 for the next 3 h together with TPA. MITF-WT protein in cytoplasmic (C) and nuclear (N) fractions were visualized using FLAG antibody. GAPDH and γH2AX were loading controls for cytoplasmic and nuclear fractions, respectively. (F) Western blot analysis of the stability of the MITF-WT and MITF-sl mutant proteins after knocking down AKIRIN2, a key regulator of the nuclear import of proteasomes, for 24 h and then inducing MITF expression using dox for 6 h. The inducible A375P cells were treated with 40 µg/ml CHX for 0, 1, 2, and 3 h. The MITF proteins were then visualized by western blot using FLAG antibody. Actin was used as a loading control. The band intensities were quantified using ImageJ software. (G) The intensities of the indicated pS73- and S73-MITF protein bands were quantified from western blot analysis in (F) with ImageJ software and are depicted as relative protein expression to DMSO. Error bars represent SEM of three independent experiments. Statistically significant differences were calculated using unpaired Student's t test. P values for the pS73-MITF form of WT and sl compared between siCTRL and siAKIRIN2 treated conditions were 0.8860 and 0.8731. P values for the S73-MITF form of WT and sl compared between siCTRL and siAKIRIN2 treated conditions were 0.0293 and 0.0395. Source data are available online for this figure.

shutting the protein out of the nucleus increases stability. Treating the cells for 3 h with MG132 in the presence of TPA revealed a significant increase of MITF, primarily in the nucleus (Fig. 4E).

Dox-inducible A375P melanoma cells expressing MITF-WT, MITF-sp, and MITF-sl were exposed to CHX and the nuclear export inhibitor leptomycin B (LMB) (Sun et al, 2013) for different time points before harvesting for Western blotting. The results showed that the stability of pS73-MITF-WT and pS73-MITF-sp was significantly reduced upon LMB treatment, whereas the stability of S73 was not changed, and the stability of both pS73- and S73-MITF-sl was decreased upon LMB treatment (Appendix Fig. S9A,B). AKIRIN2 is essential for proteasomal degradation in the nucleus (de Almeida et al, 2021). We knocked down AKIRIN2 in our dox-inducible A375P cells expressing MITF-WT and MITF-sl prior to CHX treatment. After treating the cells for 24 h with

siAKIRIN2, the expression of the mRNA AKIRIN2 was significantly decreased (Appendix Fig. S9C), and the expression of both S73-MITF-WT and S73-MITF-sl was significantly increased (Fig. 4F,G; Appendix Fig. S9D). Our results show that MITF is degraded in the nucleus through the proteasomal pathway. The increased nuclear presence of the MITF-sl protein may explain its reduced stability.

## The K316R and E318K mutations together with the S409A mutation reduce MITF stability and increase its nuclear presence

To determine if the SUMOylation site at K316 was involved in mediating MITF subcellular localization, we replaced the K316 residue with arginine in MITF-WT and MITF-sp. We also determined the effects of the E318K mutation since individuals

carrying this mutation in MITF are predisposed to melanoma, and the mutation abolishes SUMOylation at K316 (Bertolotto et al, 2011; Yokoyama et al, 2011). Alone, neither the K316R nor the E318K mutations altered the localization of the MITF-WT or MITF-sp proteins (Fig. 5A–D). However, the double mutant proteins K316R-S409A and E318K-S409A were more nuclear, regardless of the S73-phosphorylation status (Fig. 5A–D). Critically, both double mutants were able to override the effects of TPA on nuclear export, resulting in equal distribution between the nucleus and cytoplasm (Fig. 5C,D).

However, when together with the S384A, S397A, S401A, or S405A mutations, the K316R and E318K mutations did not alter the nuclear localization of MITF or nuclear export upon TPA treatment (Appendix Fig. S10A,B). Taken together, this suggests that a specific interaction between the SUMOylation site at K316 and the phosphorylation site at S409 is important for mediating MITF export. However, since these double mutants do not fully replicate the effects of the MITF-sl protein on localization, additional regions within the C-end must be important as well.

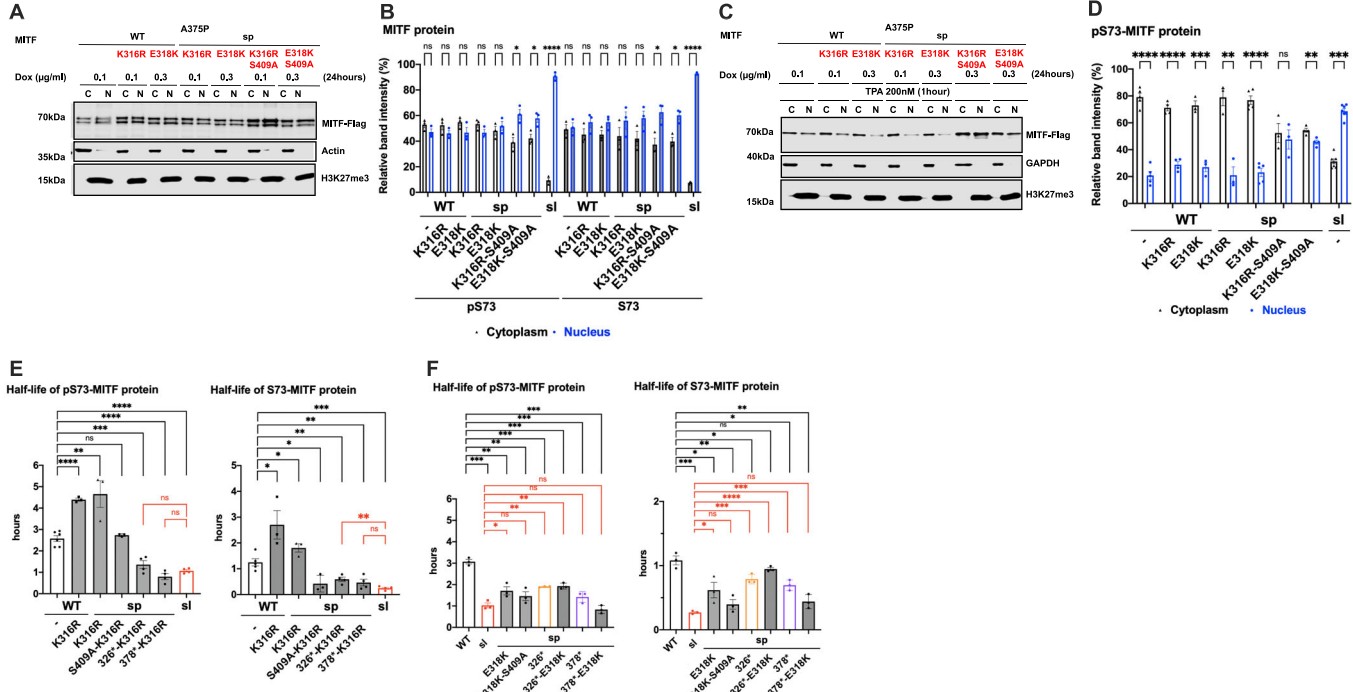

**Figure 5. The interplay between SUMOylation at K316 and phosphorylation site at S409 in regulating MITF protein stability and localization.**

(A) Western blot analysis of subcellular fractions isolated from A375P melanoma cells induced for 24 h to overexpress the indicated MITF mutant proteins. The MITF proteins in cytoplasmic (C) and nuclear (N) fractions were visualized using FLAG antibody. Actin or GAPDH and H3K27me3 were loading controls for cytoplasmic and nuclear fractions, respectively. (B) The intensities of the indicated pS73- and S73-MITF proteins in the cytoplasmic and nuclear fractions from western blot analysis in (A) were quantified separately with ImageJ software and are depicted as percentages of the total amount of protein present in the two fractions. Error bars represent SEM of three independent experiments. Statistically significant differences were calculated using unpaired Student's t test. P values for the pS73-MITF form of WT, WT-K316R, WT-E318K, sp-K316R, sp-E318K, sp-K316R-S409A, sp-E318K-S409A and sl were 0.1774, 0.2128, 0.1697, 0.1261, 0.4814, 0.0157, 0.0271 and <0.0001, respectively. P values for the S73-MITF form of WT, WT-K316R, WT-E318K, sp-K316R, sp-E318K, sp-K316R-S409A, sp-E318K-S409A and sl were 0.7878, 0.2038, 0.0958, 0.2712, 0.1098, 0.0222, 0.0101 and <0.0001, respectively. (C) Western blot analysis of subcellular fractions isolated from A375P melanoma cells induced for 24 h to overexpress the indicated MITF mutant proteins before treatment with 200 nM TPA for 1 h leading to phosphorylation of S73 of MITF. The mutant MITF proteins in cytoplasmic (C) and nuclear (N) fractions were visualized using FLAG antibody. GAPDH and H3K27me3 were loading controls for cytoplasmic and nuclear fractions, respectively. (D) The intensities of the indicated pS73-MITF proteins bands in the cytoplasmic and nuclear fractions of the western blot analysis in (C), respectively, were quantified separately with ImageJ software and are depicted as percentages of the total amount of protein present in the two fractions. Error bars represent SEM of three independent experiments. Statistically significant differences were calculated using unpaired Student's t test. P values for the pS73-MITF form of WT, WT-K316R, WT-E318K, sp-K316R, sp-E318K, sp-K316R-S409A, sp-E318K-S409A and sl were <0.0001, <0.0001, 0.0007, 0.0028, <0.0001, 0.6584, 0.0060, and <0.0001, respectively. (E, F) Half-life analysis of the pS73- and S73-MITF proteins over time after CHX treatment. The MITF protein levels relative to T0 were calculated, and non-linear regression analysis was performed. Error bars represent SEM of at least three independent experiments. Statistically significant differences were calculated using unpaired Student's t test. P values for the pS73-MITF form of WT-K316R, sp-K316R, sp-K316R-S409A, sp-326*-K316R, sp-378*-K316R, and sl compared to WT were <0.0001, 0.0026, 04614, 0.0007, <0.0001, and <0.0001, respectively. P values for the pS73-MITF form of sp-326*-K316R and sp-378*-K316R compared to sl were 0.1869 and 0.1159, respectively. P values for the S73-MITF form of WT-K316R, sp-K316R, sp-K316R-S409A, sp-326*-K316R, sp-378*-K316R, and sl compared to WT were 0.0169, 0.0452, 0.0118, 0.0072, 0.0050, and 0.0005, respectively. P values for the S73-MITF form of sp-326*-K316R and sp-378*-K316R compared to sl were 0.0093 and 0.1561, respectively. P values for the pS73-MITF form of sl, sp-E318K, sp-E318K-S409A, sp-326*, sp-326*-E318K, sp-378*, and sp-378*-E318K compared to WT were 0.0002, 0.0033, 0.0022, 0.0004, 0.0010, 0.0008, and 0.0001, respectively. P values for the pS73-MITF form of sp-E318K, sp-E318K-S409A, sp-326*, sp-326*-E318K, sp-378*, and sp-378*-E318K compared to sl were 0.0379, 0.1455, 0.0017, 0.0031, 0.1045, and 0.2853, respectively. P values for the S73-MITF form of sl, sp-E318K, sp-E318K-S409A, sp-326*, sp-326*-E318K, sp-378*, and sp-378*-E318K compared to WT were 0.0003, 0.0288, 0.0026, 0.0238, 0.1406, 0.0100, and 0.0024, respectively. P values for the S73-MITF form of sp-E318K, sp-E318K-S409A, sp-326*, sp-326*-E318K, sp-378*, and sp-378*-E318K compared to sl were 0.0453, 0.1788, 0.0004, <0.0001, 0.0010, and 0.0616, respectively. Source data are available online for this figure.

To clarify the role of the SUMOylation site on MITF protein stability, we tested the stability of the MITF-WT, MITF-sp, MITF-sp-378*, MITF-sp-326* proteins, and MITF-S409A in the presence of the K316R mutation. The pS73-MITF-WT-K316R and pS73-MITF-sp-K316R proteins were significantly more stable than pS73-MITF-WT; the stability of the S73-MITF-WT-K316R and S73-MITF-sp-K316R proteins was slightly but not significantly increased (Figs. 5E and EV5A). Interestingly, the pS73-MITF-sl protein was significantly less stable than the pS73-MITF-sp-326* protein (Fig. 3F,G), whereas the stability of MITF-sp-378*-K316R was comparable to pS73-MITF-sl (Figs. 5E and EV5A). This suggests that in the presence of residues 326–419, the K316R mutation increases the stability of MITF, whereas in its absence, K316R mimics the effects of the MITF-sl mutation and reduces stability. Furthermore, the double mutation K316R-S409A appears to have a specific effect on the stability of the S73-MITF-sp protein. However, the stability of pS73-MITF-sp forms remained unaffected by this mutation.

We also determined the stability of MITF-sp, MITF-sp-326*, MITF-sp-378*, and MITF-sp-S409A constructs carrying the E318K mutation. The stability of these proteins was significantly reduced in the presence of the E318K mutation (Figs. 5F and EV5B). The MITF-sp-378*-E318K protein showed similar stability as MITF-sl (Figs. 5F and EV5B), whereas MITF-sp-326*-E318K and MITF-sp-326* proteins were equally stable and both more stable than MITF-sl, regardless of phosphorylation at S73 (Figs. 5F and EV5B). The E318K-S409A mutation resulted in reduced stability of both pS73- and S73-MITF-sp. Taken together, our findings suggest that the E318K mutation reduces the stability of MITF. Interestingly, the K316R and E318K single mutations have different effects on stability even though both eliminate SUMOylation at K316. We hypothesize that the carboxyl domain (aa 378–419) of MITF interacts with the SUMO-site at K316, determining the stability of MITF.

### The carboxyl-end IDRs are dynamic and proximal in the dimer

To probe the interactions between the regions containing the SUMOylation site at K316 and the phosphorylation site at S409, the distance between these domains was measured using smFRET (Schuler et al, 2016). To enable site-specific labeling, recombinant human MITF constructs with only one or two native cysteines were used for inter- and intramolecular FRET, respectively. When probed intermolecularly, C306-C419 showed a high mean transfer efficiency denoted as $\langle E \rangle = 0.66$ (Fig. 6A). However, when probed intramolecularly between positions C306-C419 showed a slightly expanded chain with $\langle E \rangle = 0.30$ (Fig. 6B). This indicates that the S409 phosphorylation site of one chain is closer to the K316 SUMOylation region of its partner molecule than it is to its own K316 site. These results suggest that crosstalk between the K316 SUMOylation and S409 phosphorylation region may be between the different monomers of MITF rather than within each monomer. Analysis of the donor fluorescence lifetimes shows that distances between and within the C-terminal IDRs are dynamic on the µs timescale, suggesting the C-terminal IDRs probably do not directly interact with each other to form stable tertiary structures (Fig. 6C).

## Discussion

Suppressor mutation screens have provided valuable information about gene function, molecular pathways, and protein-protein interactions (Bautista et al, 2021; Sujatha and Chatterji, 2000). Suppressor screens are commonly performed in yeast, Drosophila, and C. elegans but rarely in mice or other mammals. Here, we generated a novel intragenic suppressor mutation at the *Mitf* locus in the mouse and showed that it is a re-mutation at the *Mitf* locus, which results in a truncation of the already mutated MITF-sp protein.

In the homozygous condition, the *Mitf^{mi-sl}* mutation leads to brownish coat color compared to the normal black coat of *Mitf^{mi-sp}* homozygotes. However, in compound heterozygous conditions with other *Mitf* mutations, including severe dominant-negative or loss-of-function mutations, the *Mitf^{mi-sl}* mutation restores the coat color phenotype compared to combinations of the same alleles with the original *Mitf^{mi-sp}* mutation. At the molecular level, this suppressor mutation increases the nuclear localization of the MITF-sl protein and reduces its stability. The "brownish" phenotype of *Mitf^{mi-sl}* homozygotes is likely to be due to the reduced stability of the MITF-sl protein and the consequent

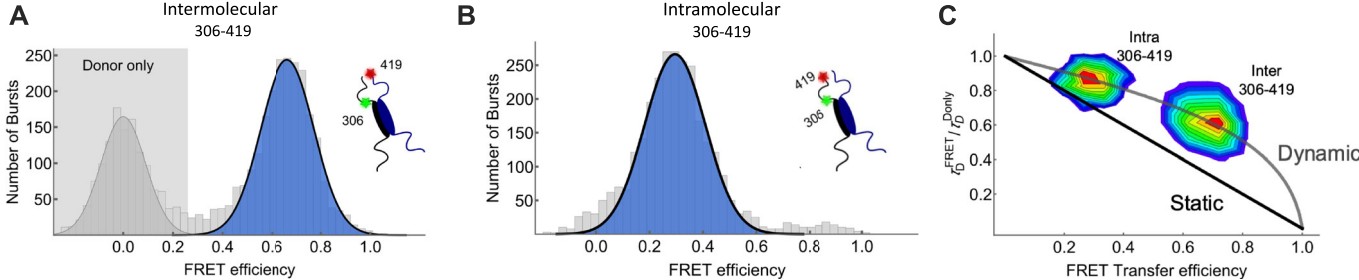

**Figure 6. Inter- and intramolecular FRET indicates MITF C-end IDRs are proximal to each other and dynamic.**

(A, B) Single-molecule Förster resonance energy transfer histograms of dimeric MITF fluorescently labeled at residues C306 and C419. Intermolecular FRET (A) showing fitted Gaussian population mean, $E = 0.66$, and intramolecular FRET (B), showing fitted Gaussian population mean, $E = 0.3$. FRET populations fitted with a Gaussian distribution are indicated in blue with donor-only events in grey. (C) Fluorescence lifetime analysis of inter- and intramolecular distances of the C-terminal IDR of MITF. The 2-D plot shows the lifetime of the donor in the presence of the acceptor ($\tau_D^{FRET}$) relative to the donor fluorescence in its absence ($\tau_D^{Donly}$), plotted against the FRET transfer efficiency for each burst. The solid black line shows the expected relationship for a static distance, while the grey line shows the expected relationship for a dynamic chain with a Gaussian distribution of distances. Source data are available online for this figure.

reduction in expression of some pigmentation genes, including *Pmel*, *Tyrp1*, and *Mlana* (Fig. EV3); (Popp et al, 2021). Interestingly, mutations in *Tyrp1* lead to mice with brown coat color (Jackson, 1988).

The total concentration of active MITF-sl protein in the nucleus at any given time will depend on the relationship between effects on nuclear import and stability. Expression of the MITF-partner proteins TFEB and TFE3 is limited in melanocytes, so they are likely to have negligible effects on MITF activity in the homozygous situation. Importantly, when the *Mitf^mi-sl* mutation is combined with any of the various other *Mitf* mutations (Figs. 1 and EV1), its ability to dimerize and translocate its partner proteins (MITF-WT or mutant MITF) into the nucleus help to explain the suppressor effects of the *Mitf^mi-sl* mutation. When in the nucleus, dimers between MITF-sl and any of the defective DNA-binding proteins MITF-Wh, MITF-mi, and MITF-ew slow down MITF-sl degradation but these dimers cannot bind DNA or activate gene expression (Hemesath et al, 1994). Eventually, however, MITF-sl monomers will be released from their non-DNA-binding dimeric partner, thus leading to the formation of MITF-sl homodimers, which can bind DNA and regulate the expression of target genes. Here, the combined effects of nuclear import, rate of nuclear degradation, DNA binding, and dimerization properties are likely to determine the final outcome; the steady-state levels of nuclear MITF-sl are likely to be determined by the rate of heterodimer dissociation and rate of degradation. The near-normal coat color phenotype of *Mitf^mi-sl*/*Mitf^mi* compound heterozygotes suggests that together these effects result in almost full MITF activity during critical stages of melanocyte development and function. This is a novel mechanism of genetic suppression and may partly explain the normal phenotypes observed in humans carrying deleterious mutations on both alleles of genes (Sulem et al, 2015).

The *Mitf^mi-sl* mutation also provides novel insights into how both stability and nuclear export of the MITF protein are regulated. Nuclear localization of MITF has been shown to involve a balance between import and export that depends on a number of domains, including a nuclear localization signal in the DNA-binding domain of MITF and an export signal that depends on the S69 and S73 phosphorylation sites (Fock et al, 2019; Ngeow et al, 2018); (Fig. 1A). In wild-type cells, MITF is approximately equally distributed between the nucleus and cytoplasm as determined by western blotting, although, due to differences in nuclear and cytoplasmic volumes, it is more concentrated in the nucleus than in the cytoplasm, as evidenced by immunocytochemistry (Fock et al, 2019). Our observations show that the C-end of MITF has major effects on nuclear localization and that residues 316–326, 336–350, and 374–419 are major factors in mediating the nuclear export of MITF. Interestingly, simultaneously mutating the SUMO-site at K316 and the phosphorylation site at S409 increased the nuclear localization of MITF compared to either single mutant alone, suggesting that these two post-translational modifications are necessary for nuclear export (Fig. 7). Importantly, the effects of the nuclear export signal mediated by the S73 and S69 phosphorylation (Ngeow et al, 2018) are less efficient when missing the C-end.

Our results show that MITF is mainly degraded through a nuclear ubiquitin-proteasomal degradation pathway. Again, the effects on stability are mainly mediated by the domains encoded by residues 316–326 and 378–419 where K316 and S409 play an

important role. However, since all our deletion constructs showed some effect, most regions within the C-domain seem to affect protein stability. This suggests that the entire domain may be important, as is often observed for IDRs. Since the MITF-sl protein is quickly degraded in the nucleus, it is likely that truncation at the C-end activates a degradation signal. A degron motif was recently discovered in the amino end of the A isoform of MITF (Nardone et al, 2023), but as this is not present in the melanocyte-specific M-isoform studied here, another degradation signal must be involved here. The fact that the effects on nuclear localization and stability are primarily encoded by the same domains suggests that these events may be related. The effects on nuclear localization are likely dominant since the protein will be degraded by the nuclear proteasome machinery if located in the nucleus. As another layer of regulation, when in the nucleus, MITF stability will also further be regulated by other factors, including SUMOylation at K316 and phosphorylation at S409; DNA binding may also be important, potentially by mediating structural changes. However, how DNA binding contributes to MITF-sl stability is not clear. Importantly our work suggests that the interaction between the SUMOylation site at K316 and the phosphorylation site at S409 is important for regulating MITF localization and stability. Our smFRET results show that these two regions are near each other in space, suggesting that direct interactions between the different protomers are involved.

In contrast to previous literature (Wu et al, 2000), our work shows that the S73 form of MITF-WT is much less stable than the pS73 form. The only explanation we have at this point for this difference is that the cell-based systems used are different. The dynamic nature of the phosphorylation/dephosphorylation and nuclear import/export processes may be different in the cell lines used. In our model, there is an almost 3-fold difference in stability between the two forms of MITF-WT. Interestingly, the *Mitf^mi-sl* mutation reduced the stability of both the pS73 and S73 forms of MITF about 3-fold in each case (Fig. 2A,B), suggesting that the effects of the C-end on stability did not influence the relative stability of the pS73 and S73 forms. However, the MITF-sl protein had a higher pS73/S73 ratio than MITF-WT, suggesting the possibility that the C-end may affect the kinetics at S73 phosphorylation. It is possible that the difference between the pS73 and S73 forms is due to continuous phosphorylation of the S73-form, possibly mediated by doxycycline treatment, thus affecting the ratio between the two forms of the protein, leading to nuclear export. The observation that pS73-MITF is exported from the nucleus (Ngeow et al, 2018) suggests that the kinetics of S73 phosphorylation and dephosphorylation may determine subcellular location and thus mediate protein stability. The enhanced accumulation of the S73 form facilitated by proteasomal inhibitor treatment (Fig. 4C) and AKIRIN2 knockdown (Fig. 4F) suggests that S73 is the stable form of MITF in the nucleus. Currently, there is limited information on the kinetics or pathways involved.

Independent reports have shown that the E318K variant in human MITF predisposes to melanoma (Bertolotto et al, 2011; Yokoyama et al, 2011). This variant alters an essential residue in the SUMOylation motif ΨKXD/E, which includes K316, the actual SUMOylation site. We show that the E318K mutant protein, which cannot be SUMOylated at this site, exhibits normal nuclear localization. However, when the S409A mutation is also present,

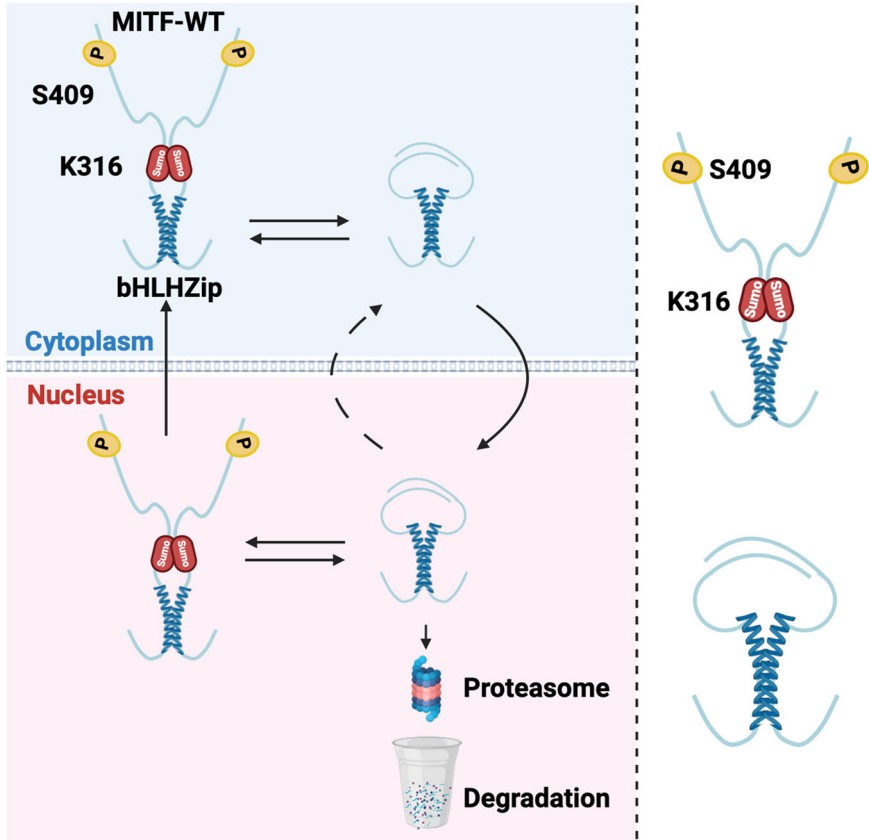

**Figure 7.** A model depicting the role of the 316–419 domain in regulating MITF stability and localization.

the protein is more nuclear, regardless of S73 phosphorylation status. The E318K mutation resulted in reduced MITF stability both in the presence and absence of the S409A mutation. S409 has been suggested to be phosphorylated by the MAP-kinase p90Rsk (Wu et al, 2000) or by AKT (Wang et al, 2016). The gain-of-function BRAF$^{V600E}$ mutation and loss-of-function PTEN mutations might accelerate the p90Rsk or AKT kinase activity, respectively, and promote S409 phosphorylation. These effects might promote cytoplasmic retention of MITF-E318K, which subsequently would increase the stability of MITF and maintain the level of MITF protein at steady-state levels leading to tumour initiation. Thus, depending on environmental signals (e.g., sun exposure), the medium-risk allele E318K (Bertolotto et al, 2011; Yokoyama et al, 2011) can mediate disease predisposition.

Based on our data, we propose a model where the two regions of the C-end of MITF, the SUMOylation site at K316 and the phosphorylation site at S409, are impacted by SUMOylation and phosphorylation, leading to effects on nuclear localization and stability. In the absence of SUMOylation at K316 and phosphorylation at S409, these residues are close in space and may collapse around the zipper domain, thus hiding nuclear export signals while at the same time exposing degradation signals (Fig. 7). However, SUMOylation and phosphorylation may change the conformation, leading to an extended version of the C-end, thus exposing nuclear export, hiding degradation signals, and affecting DNA binding. Interestingly, it has been reported that the unphosphorylated S409

MITF is required to maintain the association of MITF and PIAS3, which enables SUMOylation at K316. This may represent a feedback loop to limit the activity of MITF at any given time based on environmental signals. We, therefore, conclude that generating suppressor mutations in the mouse is an exciting and feasible option for studying gene function and may reveal unexpected aspects of protein function and regulation, leading to novel insights into protein activities in the living organism.

## Methods

### Reagents and tools table

| Reagent/resource | Reference or source | Identifier or catalog number |
|---|---|---|
| **Experimental models** | | |
| C57BL/6J (*M. musculus*) | Jackson Laboratories | N/A |
| NAW (*M. musculus*) | National Cancer Institute | N/A |
| 82UT (*M. musculus*) | Oak Ridge National Laboratories | N/A |
| C3H/C57BL/6J (*M. musculus*) | National Cancer Institute | N/A |

| Reagent/resource | Reference or source | Identifier or catalog number |
|---|---|---|
| A375P cell (*H. sapiens*) | ATCC | CRL-3224 |
| SkMel28 cell (*H. sapiens*) | ATCC | HTB-72 |
| 501Mel cell (*H. sapiens*) | Dr. Ruth Halaban (Yale University) | N/A |
| **Recombinant DNA** | | |
| pPB-hCMV1-EV-3XFLAG-HA | (Dilshat et al, 2021) | N/A |
| py-CAG-pBase | (Magnúsdóttir et al, 2013) | N/A |
| pPB-CAG-rtTA-IRES-Neo | (Dilshat et al, 2021) | N/A |
| pET24a vector | Addgene | #69749-3 |
| **Antibodies** | | |
| Anti-FLAG | Sigma | #F3165 |
| Anti-β-Actin | Cell Signaling | #4970 |
| Anti-GAPDH | Cell Signaling | #5174 |
| Anti-gH2AX | Abcam | #ab2251 |
| Anti-H3K27me3 | Cell Signaling | #9733 |
| Anti-GFP | Abcam | #ab290 |
| Anti-MITF 6A5 | (Bharti et al, 2008b) | N/A |
| C5-monoclonal MITF antibody | Abcam | #ab12039 |
| DyLight 800 anti-mouse | Cell Signaling | #5257 |
| DyLight 580 anti-rabbit IgG | Cell Signaling | #5366 |
| Rabbit IgG | Cell Signaling | #3900S |
| **Oligonucleotides and other sequence-based reagents** | | |
| Positive strand/5AmMC6/ GAGATCATGTGTTGA | IDT Inc | N/A |
| Negative strand/5AmMC6/ TCAACACATGATCTC | IDT Inc | N/A |
| Labeled probe (5'-AAAGTCAGT CATGTGCTTTTCAGA-3') | IDT Inc | N/A |
| qPCR primer | This study | N/A |
| **Chemicals, enzymes and other reagents** | | |
| RPMI 1640 medium | Gibco | #5240025 |
| FBS | Gibco | #10270–106 |
| Cycloheximide | Sigma | #66819 |
| MG132 | Sigma | #474790 |
| Doxycycline | Sigma | #324285 |
| TPA | Merck | #P1585 |
| Baf-A1 | Merk | #88899-55-2 |

| Reagent/resource | Reference or source | Identifier or catalog number |
|---|---|---|
| Leptomycin B | Merk | #L2913 |
| G418 | Gibco | #10131-035 |
| TRIzol reagent | ThermoFisher | (#15596–026 |
| HEPES | Sigma | #H3375 |
| Magnesium chloride hexahydrate | Sigma | #7791-18-6 |
| Potassium chloride | Sigma | #P3911 |
| DTT | Sigma | #3483-12-3 |
| Halt™ Protease and Phosphatase Inhibitor Cocktail, EDTA-free (100X) | ThermoFisher | #78445 |
| NP-40 Surfact-Amps™ Detergent Solution | ThermoFisher | #85124 |
| Sodium chloride | Sigma | #S5886 |
| Dynabeads Protein G magnetic beads | Invitrogen | #10004D |
| Concanavalin A-coated magnetic beads | EpiCypher | #21-1401 |
| Phosphate buffered saline | Sigma | #P4417 |
| Tween-20 | Sigma | #9005-64-5 |
| EDTA | ThermoFisher | #17892 |
| Ethylene glycol | ThermoFisher | #29810 |
| Glycerol | ThermoFisher | #15514011 |
| Bovine serum albumin (BSA) | Sigma | #A7030 |
| FuGENE HD reagent | Promega | #E2311 |
| NativePAGE Sample Buffer | Thermo Scientific | #BN2003 |
| PMSF | Sigma | #329-98-6 |
| Triton X-100 | ThermoFisher | #HFH10 |
| TCEP | Sigma | #E3889 |
| Imidazole | Sigma | #288-32-4 |
| Urea | ThermoFisher | #AM9902 |
| Spermidine | Sigma | #124-20-9 |
| RNAse | ThermoFisher | #EN0531 |
| EGTA | ThermoFisher | #67-42-5 |
| Digitonin | Sigma | #11024-24-1 |
| Proteinase K | ThermoFisher | #EO0491 |
| **Software** | | |
| Bowtie2 version 2.1.0 | (Langmead and Salzberg, 2012) | |
| MACS version 2.1 | (Zhang et al, 2008) | |

| Reagent/resource | Reference or source | Identifier or catalog number |
|---|---|---|
| ChIPseeker | (Yu et al, 2015) | |
| Diffbind R package | (Stark and Brown, 2011) | |
| GraphPad Prism 9.0 software | https://www.graphpad.com | |
| ImageJ | https://imagej.nih.gov/ij/index.html | |
| **Other** | | |
| Gibson Assembly Cloning kit | New England Biolabs | # E5510S |
| Q5 Site-directed Mutagenesis Kit | New England Biolabs | #E0554S |
| RNAwiz kit | ThermoFisher | #Am1925 |
| SuperScript reverse transcriptase | Invitrogen | #18080093 |
| High-Capacity cDNA Reverse Transcription Kit | Applied Biosystems | #4368814 |
| SensiFAST SYBR Lo-ROX Kit | Bioline | #BIO-94020 |
| NativePAGE kit | Thermo Scientific | # BN1001 |
| PVDF membranes | Thermo Scientific | #88520 |
| Illumina's NEBNext Ultra II library preparation reagents | New England Biolabs | #E7645L |

## Mouse strains used, mutagenesis and genotyping

The following *Mitf* mutants were used in this study: C57BL/6J-*Mitf*[mi-sp], C57BL/6J-*Mitf*[mi-eyeless white] (*Mitf*[mi-ew]), NAW-*Mitf*[mi-ew], C57BL/6J-*Mitf*[Mi-Wh], C57BL/6J-*Mitf*[mi-red-eyed white] (*Mitf*[mi-rw]), C57BL/6J-*Mitf*[microphthalmia] (*Mitf*[mi]), 82UT-*Mitf*[mi-Oak ridge] (*Mitf*[Mi-Or]), C57BL/6J-*Mitf*[Mi-or], 82UT-*Mitf*[mi-brownish] (*Mitf*[mi-b]) and a [C3H/C57BL/6 J]-*Mitf*[mi-vga-9] (Table 1). The *Mitf*[mi-ew] mutation arose on the NAW background (Miner, 1968), and the *Mitf*[Mi-Or] mutation on the 82UT background (Stelzner, 1964) both were subsequently backcrossed for 10 generations to C57BL/6J. To screen for Mitf suppressor mutations, homozygous B6-*Mitf*[mi-sp] males were treated four times at 1-week intervals with 100 mg/kg ENU. After a 6–8-week recovery period, the males were mated to NAW-*Mitf*[mi-ew]/*Mitf*[mi-ew] females. The resulting offspring, which all should show an identical phenotype, were screened for abnormally pigmented deviates. DNA-HPLC was used to confirm the presence of the *Mitf*[mi-sp] mutation. Mice in Iceland were maintained in accordance with Icelandic law number 55/2013 and Regulation number 460/2017 both of which are in accordance with European Union Directive 2010/63. A licence was issued by the Committee on Experimental Animals in May 2019 and renewed in March 2023 (licence 2301531). Mice at the NIH were bred and maintained in a pathogen free facility certified by the Association for Assessment and Accreditation of Laboratory Animal Care (AAALAC) International, and the study was carried out in accordance with protocols approved by the NCI Frederick Animal Care and Use Committee (ACUC). Animal care was provided in accordance with the procedures outlined in the "Guide for Care and Use of Laboratory Animals" (National Research Council; 2011; National Academies Press; Washington, D.C.). Animals were kept in the NCI vivarium in colony cages at an ambient temperature of $25 \pm 2\,°C$ and 45–55% relative humidity with 12 h light:dark cycle. They had free access to standard rodent pellet diet and drinking water.

## Cell culture, reagents, and antibodies

Melanocyte cultures were obtained from trypsin-digested P0-P3 skin using microscopic selection of pigmented cells. The cell lines A375P (CRL-3224), and SkMel28 (HTB-72) were purchased from ATCC. 501Mel melanoma cells were obtained from the lab of Dr. Ruth Halaban (Yale University). The cells were maintained in RPMI 1640 medium (Gibco, #5240025) supplemented with 10% FBS (Gibco #10270–106) at 5% $CO_2$ and 37 °C in a humidified incubator. Cycloheximide (CHX – 50 mg/ml) (Sigma, #66819), 20 mg/ml MG132 (Sigma, #474790), 10 mg/ml Doxycycline (Dox – Sigma, #324285), 10 mg/ml TPA (Merck, #P1585), 1 mg/ml Baf-A1 (Merk, #88899-55-2), 5 mM Leptomycin B (Merk, #L2913), stock solutions were prepared in DMSO. The primary antibodies used for all western blot (WB) experiments and their dilutions were as follows: Anti-FLAG (Sigma, #F3165) at 1:5000 dilution; Anti-β-Actin (Cell Signaling, #4970) at 1:1000 dilution; Anti-γH2AX (Abcam, #ab2251) at 1:2000 dilution, Anti-GAPDH (Cell Signaling, #5174), at 1:1000 dilution; Anti-H3K27me3 (Cell Signaling, #9733) at 1:1000 dilution; Anti-GFP (Abcam, #ab290) at 1:2500 dilution.

## Generation of plasmid constructs for stable doxycycline-inducible overexpression

Fusions of wild-type and mutant mouse MITF-M cDNA with the 3XFLAG-HA tag at the C- or N-terminus or fusion with the GFP-tag at the C terminus were generated in the piggy-bac vector pPB-hCMV1. The cDNAs were subcloned downstream of a tetracycline response element (TRE) using the Gibson Assembly Cloning kit (New England Biolabs, # E5510S). Mutations were introduced by in vitro mutagenesis using Q5 Site-directed Mutagenesis Kit (New England Biolabs, #E0554S) according to the manufacturer's instructions.

## Generation of stable doxycycline-inducible MITF

Inducible A375P, SKmel28, and 501Mel cells were generated as described before (Dilshat et al, 2021). Briefly, the wild-type and mutant mouse MITF fusion constructs with the 3XFLAG-HA at the C- or N-terminus or fusion with GFP at C-terminus or a pPB-hCMV1-EV-3XFLAG-HA empty vector was transfected into 70–80% confluent A735P, 501Mel or SKMel28 cells using Fugene HD reagent (Promega, #E2311) together with the py-CAG-pBase and pPB-CAG-rtTA-IRES-Neo plasmids at a 10:10:1 ratio. After 48 h of transfection, the transfected cell lines were selected with 0.5 mg/ml of G418 (Gibco, #10131-035) for 2 weeks. A 'mock plate' of no transfected cells was also included in each case. To equalize the expression of MITF proteins, the dox-inducible A375P, 501Mel, and SKmel28 melanoma cell lines were treated with varying concentrations of doxycycline, and the doxycycline concentrations leading to similar MITF protein levels were used for future experiments.

## qRT-PCR and sequencing

Total RNA was isolated from hearts of wild-type and mutant mice using the RNAwiz kit (ThermoFisher, #Am1925). The RNA was

reverse transcribed by SuperScript reverse transcriptase (Invitrogen), and the resulting cDNA phenol/chloroform was extracted. Alternatively, RNA was isolated using the Macherey Nagel RNAII kit. The entire *Mitf* cDNA was amplified by PCR using overlapping primers. The resulting PCR products were sequenced directly using the Big Dye Terminator Cycle Sequencing kit (ABI) and the ABI 377 sequencer. The results were confirmed by sequencing additional animals as well as several control animals, on which the mutation was induced, in order to confirm the alterations.

The day before inducing MITF expression by doxycycline, cells were seeded on 12-well plates at a density of $12 \times 10^4$ cells per well. MITF expression was induced for 6, 12, 24, and 36 h and harvested for RNA isolation using TRIzol reagent (ThermoFisher, #15596–026). High-Capacity cDNA Reverse Transcription Kit (Applied Biosystems, #4368814) was used for cDNA synthesis according to the manufacturer's instructions. The SensiFAST SYBR Lo-ROX Kit (Bioline, #BIO-94020) was utilized for the qRT-PCR. qRT-PCR reactions were performed using 0.4 ng/μl cDNA in triplicates. The relative fold change in gene expression was calculated using the D-$\Delta\Delta$Ct method (Livak and Schmittgen, 2001). The geometrical mean of β-actin and hARP expression was used to normalize the target gene expression.

## Subcellular fractionations

The day before inducing MITF expression by doxycycline, cells were seeded on 6-well plates at a density of $3.5 \times 10^5$ cells per well for 24 h, after which the cells were either directly harvested or treated with TPA at 200 nM for 1 h, 4 h, or treated with TPA at 200 nM for 1 h and MG132 40 μg/ml for the next 3 h in the presence of TPA before harvesting by trypsinization. Cells were washed with PBS before washing twice with swelling buffer (10 mM HEPES, pH 7.9, 1.5 mM MgCl$_2$, 10 mM KCl, 0.5 mM DTT, and freshly added protease and phosphatase inhibitors). The cells were then lysed by incubation at 4 °C for 15 min in cell lysis buffer (10 mM HEPES, pH 7.9, 1.5 mM MgCl$_2$, 10 mM KCl, 0.5 mM DTT, 0.1% NP40). Approximately 30% of the sample was collected and set aside as whole cell lysate. The remaining cell lysate was spun down at 3000 rpm for 5 min at 4 °C, and the supernatant was collected as the cytoplasmic fraction. At the same time, the pellet, representing the nuclear fraction, was washed with cold PBS before resuspension in RIPA buffer (20 mM Tris-HCl, pH 7.4, 50 mM NaCl, 2 mM MgCl$_2$, 1% (v/v) NP40, 0.5% (m/v) sodium deoxycholate, and 0.1% (m/v) sodium dodecyl sulfate, and freshly added protease and phosphatase inhibitors) for further experiments including Western blotting and immunoprecipitation.

## Immunoprecipitation

Cells were seeded on 6-well plates at a density of $3.5 \times 10^5$ cells per well the day before transfection. The following day, FuGENE HD reagent (Promega # E2311) was used to conduct the co-transfection of MITF-WT, MITF$^{mi-sp}$ or MITF$^{mi-sl}$-GFP-tagged constructs together with MITF$^{mi}$, MITF$^{mi-ew}$, MITF$^{Mi-Wh}$, 14-3-3-Epsilon or 14-3-3-zeta Flag-tagged proteins. After 24 h, the cells were washed twice with ice-cold PBS and lysed by adding 200 μl of RIPA buffer with freshly added protease and phosphatase inhibitors. The cell

lysate was then ready for immunoprecipitation (IP); 30% of the sample was collected as an input fraction. For each IP sample, 20 μl of Dynabeads Protein G magnetic beads (Invitrogen, # 10004D) were washed twice with 1 ml PBS using a magnetic stand before resuspending in 300 μl of PBS containing 0.01% Tween-20. The magnetic beads were then conjugated with anti-FLAG antibodies by adding 1 μg anti-FLAG antibody (Sigma, #F3165), followed by a 30-min incubation at RT with rotation. The magnetic beads were washed twice with PBS containing 0.01% Tween-20 to eliminate non-conjugated antibodies and then resuspended with 20 μl of PBS containing 0.01% Tween-20. The IP samples were incubated with the coated beads overnight at 4 °C with rotation. Samples were then placed on the magnetic stand, and supernatants were removed and saved as an unbound fraction (UnB) in each case. The beads were washed twice with 1 ml PBS containing 0.01% Tween-20. The protein was eluted from the beads by incubating with 150 ng/μl 3× Flag peptide in PBS containing 0.01% Tween-20 for 30 min at 4 °C with rotation. The samples were placed on the magnetic stand, and supernatants were saved as an immunoprecipitation fraction (IP). The collected fractions were then subjected to western blot analysis.

## EMSA DNA binding studies

Electrophoretic Mobility Shift Assays were performed using proteins expressed in the TNT T7 Coupled Reticulocyte Lysate System (Promega, WI), according to the manufacturer's recommendation. DNA binding reactions were performed in 10 mM Hepes (pH 7.9), 50 mM NaCl, 5 mM MgCl$_2$, 0.1 mM EDTA, 2 mM dithiothreitol (DTT), 5% ethylene glycol, and 5% glycerol. Ten μL of 2X buffer were combined with 2.5 μL of TNT translated MITF protein, 1.4 ng of labeled probe (5'-AAAGTCAGTCATGTGCTTTTCAGA-3'), 4 μg of bovine serum albumin (BSA) (Lee et al, 2000) and water, adjusting the reaction volume to 20 μL. For supershifts, 0.5 μL of C5-monoclonal MITF antibody (Neomarkers) was added to the reaction. The samples were incubated on ice for 30 min to allow binding to proceed. The resulting DNA-protein complexes were resolved on 6% non-denaturing polyacrylamide gels, placed on a storage phosphor screen, and scanned on a Typhoon Phosphor Imager 8610 (Molecular Dynamics) for analysis.

## Protein degradation assay

The dox-inducible A375P cells were treated with doxycycline to express MITF WT and MITF mutant proteins for 24 h and then treated with 40 μg/ml cycloheximide (Sigma #66819), in presence or absence of 200 nM TPA, or 5 nM LMB for 0, 1, 2, and 3 h before harvesting. For protein degradation pathway analysis, the dox-inducible A375P cells were treated with doxycycline to express the respective MITF constructs for 24 h at density of $8 \times 10^4$ cells and then treated with either 20 μg/ml MG132 or 0.2 μg/ml Baf-A1 in the presence or absence of 40 μg/ml CHX for 3 h before harvesting. FuGENE HD reagent (Promega, #E2311) was used for the co-transfection. After 24 h, the cells were treated with 55 μg/ml cycloheximide (Sigma, #66819) for 0, 1, 2, and 3 h. The cells were finally lysed in SDS sample buffer (2% SDS, 5% 2-mercaptoethanol, 10% glycerol, 63 mM Tris-HCl, 0.0025% bromophenol blue, pH 6.8), and the expression of the MITF protein determined by Western blot using FLAG antibodies.

## Western blot analysis

For the Blue native PAGE, the A375P melanoma cells transiently co-expressing the MITF[mi]-Flag protein with either MITF-WT-EGFP, MITF[mi-sp]-EGFP, or MITF[mi-sl]-EGFP proteins or expressing MITF-WT-Flag, MITF[mi-sp]-Flag, MITF[mi-sl]-Flag, MITF[mi]-Flag for 24 h before harvesting in NativePAGE Sample Buffer (Thermo Scientific, #BN2003). We then performed Blue native PAGE (Wittig et al, 2006) according to the manufacturer's instructions (Thermo Scientific, # BN1001) followed by a second dimension of SDS-PAGE. For cell lysates in SDS sample buffer, the samples were boiled for 5 min at 95 °C. Proteins were then transferred to 0.2 μm PVDF membranes (Thermo Scientific, #88520). The membranes were blocked in T-TBS (20 mM Tris, pH 7.4; 150 mM NaCl; 0.01% Tween-20) containing 5% BSA. The membranes were probed with specific primary antibodies. After washing three times with T-TBS for 10 min each, the membrane was then incubated for 1 h at room temperature with either DyLight 800 anti-mouse (Cell Signaling, #5257) or DyLight 580 anti-rabbit IgG (Cell Signaling, #5366) secondary antibodies (Cell Signaling Technology). The protein bands were detected using Odyssey CLx Imager (LICOR Biosciences) and Image Studio version 2.0. The band intensity was quantified using the open-access ImageJ software (https://imagej.nih.gov/ij/).

## Recombinant protein cloning

A bacterial codon-optimized human MITF-M + ORF synthesized by GenScript Biotech was cloned using Gibson assembly (New England Biolabs, # E5510S) into a pET24a vector, C-terminally in frame with a 6xHistidine tagged SMT3 (SUMO) fusion protein. To obtain single cysteine mutant version of MITF for maleimide labeling, two successive rounds of cysteine to serine mutagenesis were performed using the QuikChange Lightning Multi Site-directed Mutagenesis kit (Agilent Technologies #210513/#210515) according to manufacturer's instructions at 1/4th scale.

## Recombinant protein expression, purification, and labeling

The protein was expressed from Lemo21(DE3) E. coli by autoinduction at 37 °C for 8–12 h (Studier, 2005). Harvested cells were resuspended in 50 mM Tris, 10 mM EDTA, 1 mM PMSF, 1% Triton X-100, pH = 8.2, lysed by sonication, and centrifuged at $40,000 \times g$ for 30 min. The insoluble material was washed in 50 mM Tris, 150 mM NaCl, 0.05% Triton X-100, pH = 8, and re-centrifuged, followed by resuspension in 8 M Urea, 500 mM NaCl, 20 mM Imidazole, 20 mM $Na_2HPO_4$, 0.1 mM PMSF, 0.1 mM TCEP, 0.01% Tween-20, pH = 8.2. The resuspended pellet was centrifuged at $40,000 \times g$ for 30 min and the supernatant was loaded onto a 5 mL HisTrap HP column (Cytiva) connected to an Äkta Pure chromatography system, using a flow rate of 5 mL/min. The column was washed with a buffer containing 6 M Urea, 1 M NaCl, 25 mM Imidazole, 20 mM $Na_2HPO_4$, 0.01% Tween-20, pH = 8.2. The column was washed with at least 4 CV of running buffer containing 5 M Urea, 500 mM NaCl, 20 mM Imidazole, 20 mM $Na_2HPO_4$, 0.01% Tween-20, pH = 8.2, and the protein eluted in 4 CV of elution buffer containing 2 M Urea, 150 mM NaCl, 300 mM Imidazole, 20 mM $Na_2HPO_4$, 0.01% Tween-20, pH = 7.8. Fractions containing more than 4 mg/mL of protein were pooled, treated with 10 mM of DTT at RT for

15 min. Eluate was diluted with a buffer containing 50 mM Tris, 150 mM NaCl, 0.05% Triton X-100, pH = 8 to a urea concentration below 2 M and protein concentration below 1 mg/mL. The His6-SUMO tag was enzymatically removed by addition of ULP-1 protease at a substrate-to-enzyme molar ratio of 1:200. The reaction was incubated at RT for 12–36 h, followed by SDS-PAGE to confirm digestion and purity. The sample was precipitated with 10% TCA, the precipitate pelleted by centrifugation, washed with EtOH and resuspended in a minimal volume of 6 M Urea, 50 mM $K_2HPO_4/KH_2PO_4$, 10 mM DTT, pH = 8.2, incubated for 1 h and centrifuged at $17,000 \times g$ for 5 min followed by application to a HiTrap Desalting column equilibrated in the same buffer containing no DTT.

For labeling, 1 mg of eluted protein at a concentration of 50–150 μM was incubated with a 2× molar excess of CF660R (Biotium) or Cy3B (Cytiva) maleimide overnight at RT. Protein was precipitated by adding of EtOH and resuspended in 7 M GdmHCl, 100 mM DTT, 50 mM $K_2HPO_4/KH_2PO_4$, pH = 6.7. Labeling steps were omitted for the MITF-WT, MITF-sp and MITF-sl constructs used for DNA binding experiments. Samples were purified using RP-HPLC on a Zorbax 300SB-C3 column, and fractions corresponding to the target protein were lyophilized. Fractions were resuspended in 7 M GdmHCl, 50 mM $K_2HPO_4/KH_2PO_4$, pH = 6.7, and analyzed by UV–Vis to determine protein concentration and degree of labeling before long-term storage at −80 °C.

## DNA probe labeling and purification

Functionalized DNA oligonucleotides were synthesized by IDT Inc, positive strand /5AmMC6/GAGATCATGTGTTGA and negative strand /5AmMC6/TCAACACATGATCTC. Both oligonucleotides were resuspended in 100 mM sodium bicarbonate to a concentration of 1 mM. The positive strand was incubated with tenfold molar excess of Cy3B NHS ester overnight at RT and the negative strand was incubated in the same way with CF660R NHS ester. Reactions were precipitated by adding 0.1 volumes of 4 M sodium acetate and 3 volumes 75% EtOH, incubated at −20 °C for an hour, followed by centrifugation at $25,000 \times g$ for 30 min. The pellet was resuspended in a minimal volume of DNA RP-HPLC solution A (100 mM triethylammonium acetate, pH = 8, 5% acetonitrile) and purified using RP-HPLC on a ReproSilGold 200å C18 column with a linear gradient of 5% RP-HPLC solution B (acetonitrile) to 55% over 35 min at 1 mL/min. Fractions corresponding to labeled oligonucleotides were collected, freeze-dried, and resuspended in DNA annealing buffer (10 mM Tris, 1 mM EDTA, 50 mM NaCl, pH = 8), mixed 1:1 at a concentration of 20 μM and annealed by incubation at 95 °C for 5 min and passive cooling to RT.

## Single-molecule fluorescence spectroscopy

Single-molecule experiments were all performed at 20–23 °C using a confocal MicroTime 200 instrument (PicoQuant). The donor dye was excited using a pulsed 513 nm diode laser (LDH-D-C-520, PicoQuant) at 20–30 μW of power (measured after the dichroic) using pulsed interleaved excitation at 20 MHz repetition rate 61. The acceptor dye was excited using a 640 nm diode laser (LDH-D-C-640, PicoQuant) at 15–20 μW of power. Excitation and emission light were collected through a 60× water-immersion objective (UPLSAPO60XW, Olympus) and focused onto a 100 or 50 μm

pinhole, separated by polarization, donor, and acceptor emission wavelengths. Photons were detected using four identical SPADs (SPCM-AQRG-TR, Excelitas Technologies).

For intermolecular FRET measurements, pairs of single-labeled hMITF-M+ (hMITF-WT) were mixed together in 7 M GdmHCl to a total concentration of 10 μM or greater at an acceptor-to-donor ratio of 5:1. The sample was then diluted to 100–400 nM in 10 mM Tris, 0.1 mM EDTA, 165 mM KCl, 0.01% Tween-20, pH = 7.4, prior to dilution to 0.5–2 nM for experiment. All experiments using single-labeled pairs were performed using ibidi μ-slide sample chambers at a volume of 40–50 μL in 10 mM Tris, 0.1 mM EDTA, 165 mM KCl, 1 mM MgCl₂, 143 mM 2-ME, 0.01% Tween-20, pH = 7.4.

For intramolecular FRET measurements, double-labeled h-MITF-M+ was diluted to a concentration of 100 nM in 10 mM Tris, 0.1 mM EDTA, 165 mM KCl, 1 mM MgCl₂, 0.001% Tween-20, pH = 7.4, prior to dilution to 200 pM or less for the experiment in the same buffer with 143 mM 2-ME added.

DNA-binding experiments were performed using a labeled M-Box probe, prepared as described above, at a concentration of 0.5 nM in 10 mM Tris, 0.1 mM EDTA, 165 or 300 mM KCl, 1 mM MgCl₂, 143 mM 2-ME, 0.01% Tween-20, 0.1 mg/mL BSA pH = 7.4. For refolding, purified hMITF-WT, hMITF-sp, and hMITF-sl were diluted 1000 to 100-fold to 500 nM in the same buffer containing 14 mM 2-ME, incubated on ice for at least 30 min and added to the M-box probe at least 15 min before measurement to ensure equilibrium conditions.

## Transfer efficiency histograms

Aggregates were removed from the raw trace by removing all 2 s bins containing more than the mean number of photons + 6 SD. Bursts were called using the delta-T method in Fretica, using a dT of 100 μs or less, and a cut minimum off of 30 photons per burst and a maximum of 1000. FRET efficiency was calculated as $E = n'_A / (n'_A + n'_D)$, where $n'_A$ and $n'_D$ are the number of acceptor and donor photons in each burst after correction for background, channel crosstalk, relative dye detection efficiency and direct acceptor excitation. Bursts asymmetric for emission after donor excitation (indicating bleaching) were filtered by removing bursts were the asymmetry confidence level,

$$\sigma = \frac{T}{2\sqrt{3}}\sqrt{1/N_D + 1/N_A},$$ exceeds 1.5. Bursts were filtered for stoichiometry, defined as $S = \frac{n_{DA} + n_{DD} - \delta n_{AA}}{n_{DA} + n_{DD} + \gamma_{PIE} n_{AA} - \delta n_A}$, where $n_{xy}$ indicates excitation source and detection channel, $\gamma_{PIE}$ is a correction factor for the relative excitation and emission intensities of the colours and $\delta = \gamma_{PIE} \frac{\alpha}{1-\alpha}$, and α is the correction factor for direct acceptor excitation (0.042). Bursts with a stoichiometry around 0.5, indicating a stoichiometric FRET pair, were selected and further filtered for bursts with high-confidence symmetry of acceptor emission after acceptor excitation and donor emission after donor excitation, in a similar manner as for the donor excitation only. Mean FRET efficiencies were found by fitting one or two Gaussian distributions to the data, with one fixed at around 0 efficiency to account for residual donor-only population.

## Fluorescence lifetime analysis

Donor lifetime in presence of acceptor, $\tau_{DA}$, was estimated from mean donor detection times, $\langle t_D \rangle$, of each burst. To analyze the FRET active population for dynamics the lifetime is normalized to the intrinsic donor lifetime $\tau_D$ obtained from the donor-only population and plotted against $E$. If the distance between the donor and acceptor is static, the relationship is $\tau_{DA}/\langle t_D \rangle = 1 - E$, but in the case of rapid distance dynamics it deviates to $\tau_{DA}/\langle t_D \rangle = 1 - E + \frac{\sigma^2}{1 - \langle E \rangle}$, where $\sigma^2 = \langle E \rangle^2 - \langle E \rangle^2 = \int_0^\infty dr[E(r) - \langle E \rangle]^2 P(r)$, and $P(r)$ is the probability density function of the distance between the dyes, $r$.

## Fluorescence correlation spectroscopy

Relative diffusion times of bound and unbound 16 bp MBox DNA was determined by cross-correlation of donor and acceptor signal fluctuations using $G_{ij}(\tau) = \frac{\langle \delta n_i(0) \delta n_j(\tau) \rangle}{\langle n_i \rangle^2}$, the resulting correlation curve was fit for a single diffusion time and a triplet state using:

$$G_{ij}(\tau) = 1 + \frac{a\left(1 + nT_{ij}e^{-\frac{\tau}{\tau_T}}\right)}{\left(\frac{1+\tau}{\tau_D}\right) + \left(\frac{1+\tau}{s^2\tau_D}\right)^{1/2}}$$

Where $a$ is the amplitude, $nT_{ij}$ and $\tau_T$ relate to the triplet and $\tau_D$, the diffusion time.

## Fluorescence anisotropy

Anisotropy was calculated for every burst in the FRET active population and fit using a Gaussian to obtain a mean anisotropy for the population at a given measurement point.

## Curve fitting

Bound populations were calculated from normalized FRET shift, anisotropy shift and FCS shift data. We assumed 2:1 binding and fit the resultant binding isotherm using:

$$\theta = \frac{c_{DNA,total} + K_D + c_{MITF,total} - \sqrt{\left(c_{DNA,total} + c_{MITF,total} + K_D\right)^2 - 4 \times c_{DNA,total} \times c_{MITF,total}}}{2 \times c_{DNA,total}}$$

## Cleavage under targets and release using nuclease CUT&RUN

Experiments were performed as previously described (Skene and Henikoff, 2017; Zhu et al, 2019) with minor modifications. A375P cells were induced to express FLAG-tagged MITF mutant fusion proteins (MITF-WT and MITF-sl) for 48 h. In total, $5 \times 10^5$ cells were then harvested using scrapers, centrifuged at 600 g for 5 min, and washed twice with washing buffer (20 mM HEPES pH 7.5, 150 mM NaCl, 0.5 mM spermidine, and 1X protease inhibitor cocktail). Afterward, cells were immobilized on pre-activated Concanavalin A-coated magnetic beads (EpiCypher #Cat 21-1401) at 4 °C for 10 min. Incubation with anti-Flag (Sigma,

#F3165) or Rabbit IgG (Cell Signaling, #3900S) in antibody buffer (washing buffer supplemented with 2 mM EDTA and 0.025% digitonin) followed, held overnight at 4 °C with rotation. The following day, cells were washed with a washing buffer containing 0.025% digitonin. Subsequently, pAG-MNase enzyme (500 µg/mL) was introduced, and $CaCl_2$ at 2 mM final concentration was added to activate the enzyme at 0 °C for 30 min. The reaction was terminated using 2X Stop buffer (340 mM NaCl, 20 mM EDTA, 4 mM EGTA, 0.02% Digitonin, 100 µg/mL RNAse A, 50 µg/mL Glycogen, and 500 pg/mL *E. coli* spike-in control). The Protein-DNA complex was released by undergoing incubation with the Proteinase K (1 µL/mL, ThermoFisher Scientific, #EO0491) for 1 h at 50 °C. DNA fragment purification utilized the Genomic DNA Clean & Concentrator kit (ZymoResearch, #D465). CUT&RUN experiments were conducted in triplicate. CUT&RUN library preparation using Illumina's NEBNext Ultra II library preparation reagents (E7645L).

Bowtie2 version 2.1.0 (Langmead and Salzberg, 2012) was utilized for read mapping against the hg38 genome assembly, employing settings such as --local --very-sensitive-local --no-unal --no-mixed --no-discordant --dovetail --phred33 -I 10 -X 700 -p 10. For peak calling from the BAM file, MACS version 2.1 (Zhang et al, 2008) was applied with the narrowPeak configuration and a p-value cutoff of 1e-5. Peaks were annotated to genes using the R package ChIPseeker (Yu et al, 2015). Differential binding analysis between the two groups was carried out using the Diffbind R package (Stark and Brown, 2011).

## Statistical analysis

Data were analyzed with GraphPad Prism 9.0 software (San Diego, CA). The results were obtained from at least three biological replicates. An unpaired *t* test was conducted to compare the two groups. A significant difference was established with *$P < 0.05$, *$P < 0.05$, **$P < 0.01$, ***$P < 0.001$, ****$P < 0.00$, and ns not significant. All the data were expressed as mean ± SEM. All single-molecule data were analyzed using Mathematica 12 and the Fretica plugin (https://schuler.bioc.uzh.ch/programs/) and error bars obtained from 2 SD of a Gaussian fit to the data.

## Data availability

The raw data of the CUT&RUN are available in GEO database under accession number GSE267956. The datasets produced in this study are available in the following databases: https://www.ncbi.nlm.nih.gov/geo/query/acc.cgi?acc=GSE226956.

The source data of this paper are collected in the following database record: biostudies:S-SCDT-10_1038-S44319-024-00225-3.

## Peer review information

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

## Acknowledgements

We thank Joanne Dietz, Fran Dorsey, Latasha Crawford, Melanie Gasper, and Anjalie Parekh and the NINDS Animal Health and Care Section for maintaining mutant mouse lines, Linda S. Cleveland, Susan Skuntz, Christian Praetorius, Bryndís K Gísladóttir and Aðalheiður G. Hansdóttir for expert technical assistance, and the NINDS sequencing facility for support. We also thank Colin Goding and Margrét Helga Ögmundsdóttir for their critical comments on the manuscript. This work was supported by grant 217768 from the Icelandic Research Fund (ES), the Icelandic Cancer Society (POH), and the University of Iceland Ph.D. Student Fund, the intramural research program of the NIH, NCI, and NINDS, and by Institut Curie.

## Author contributions

**Hong Nhung Vu**: Resources; Data curation; Formal analysis; Methodology; Writing—original draft; Project administration; Writing—review and editing. **Matti Már Valdimarsson**: Data curation; Formal analysis; Methodology; Writing—original draft; Writing—review and editing. **Sara Sigurbjörnsdóttir**: Supervision; Visualization; Writing—original draft. **Kristín Bergsteinsdóttir**: Data curation; Formal analysis; Methodology. **Julien Debbache**: Data curation; Formal analysis; Methodology; Writing—original draft. **Keren Bismuth**: Data curation; Formal analysis; Methodology; Writing—original draft. **Deborah A Swing**: Data curation; Formal analysis; Methodology. **Jón H Hallsson**: Data curation; Formal analysis; Methodology. **Lionel Larue**: Supervision; Visualization; Writing—original draft; Writing—review and editing. **Heinz Arnheiter**: Data curation; Formal analysis; Visualization; Methodology; Writing—original draft; Writing—review and editing. **Neal G Copeland**: Conceptualization; Supervision; Funding acquisition; Validation; Methodology; Writing—original draft. **Nancy A Jenkins**: Conceptualization; Supervision; Funding acquisition; Validation; Methodology; Writing—original draft. **Petur O Heidarsson**: Data curation; Formal analysis; Supervision; Visualization; Writing—original draft; Writing—review and editing. **Eiríkur Steingrímsson**: Conceptualization; Resources; Data curation; Software; Formal analysis; Supervision; Funding acquisition; Validation; Investigation; Visualization; Methodology; Writing—original draft; Project administration; Writing—review and editing.

Source data underlying figure panels in this paper may have individual authorship assigned. Where available, figure panel/source data authorship is listed in the following database record: biostudies:S-SCDT-10_1038-S44319-024-00225-3.

## Disclosure and competing interests statement

The authors declare no competing interests.

# Expanded View Figures

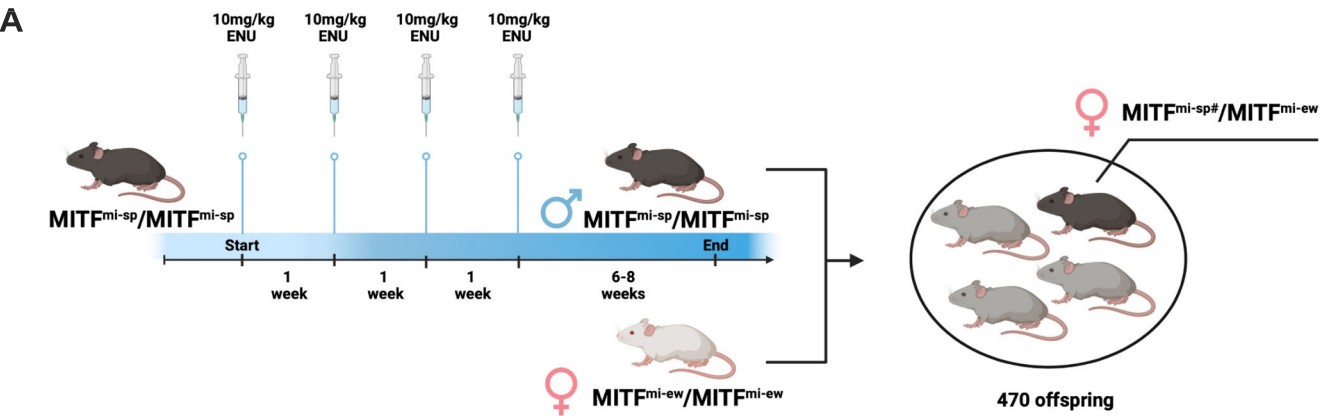

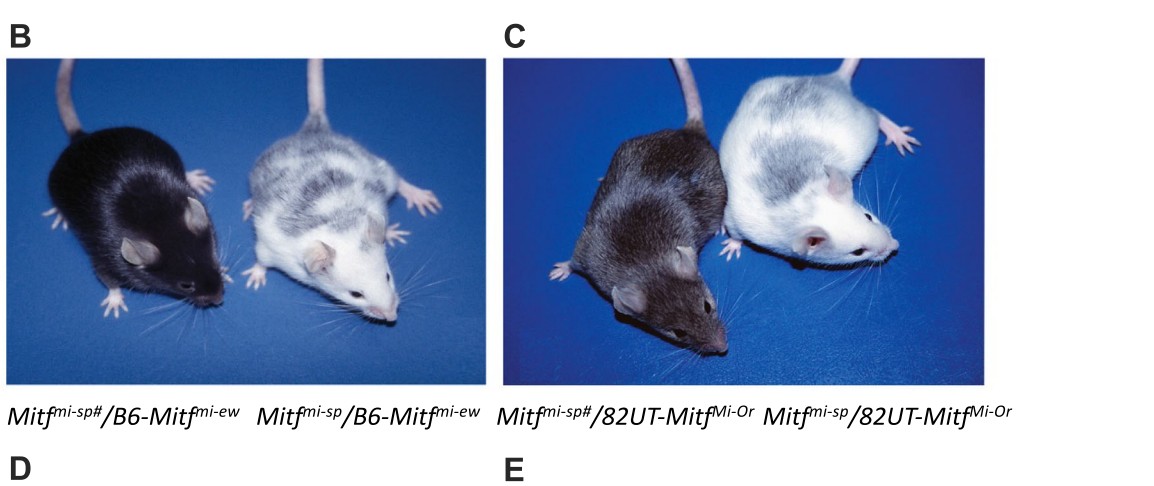

Mitf^mi-sp#/B6-Mitf^mi-ew    Mitf^mi-sp/B6-Mitf^mi-ew    Mitf^mi-sp#/82UT-Mitf^Mi-Or    Mitf^mi-sp/82UT-Mitf^Mi-Or

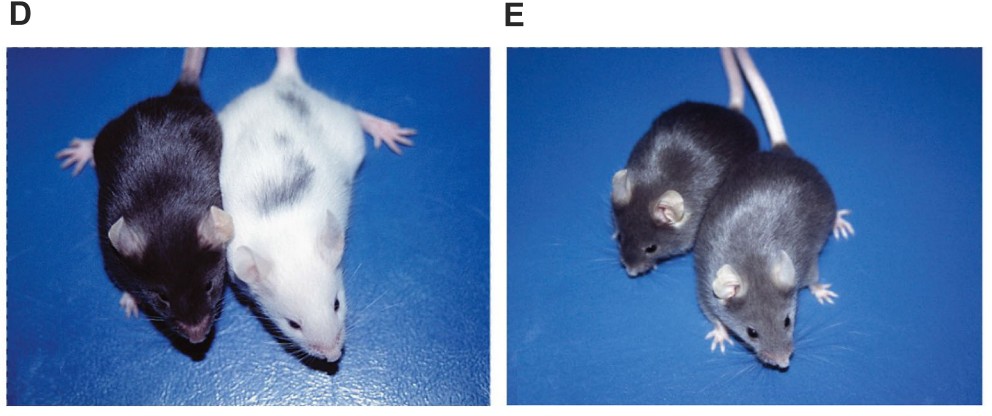

Mitf^mi-sp#/B6-Mitf^Mi-Or    Mitf^mi-sp/B6-Mitf^Mi-Or    Mitf^mi-sp#/Mitf^Mi-b    Mitf^mi-sp/Mitf^Mi-b

**Figure EV1. Generation and phenotypic behavior of the induced *Mitf^mi-sp#* suppressor mutation.**

(A) Schematic of generation of a *Mitf* suppressor mutation in mouse. (B) B6-*Mitf^mi-ew*/B6-*Mitf^mi-sp#* and B6-*Mitf^mi-ew*/B6-*Mitf^mi-sp* compound heterozygotes. (C) 82UT-*Mitf^Mi-Or*/B6-*Mitf^mi-sp#* and 82UT-*Mitf^Mi-Or*/B6-*Mitf^mi-sp* compound heterozygotes. (D) B6-*Mitf^mi-sp#*/B6-*Mitf^Mi-Or* and B6-*Mitf^mi-sp*/B6-*Mitf^Mi-Or* compound heterozygotes. (E) B6-*Mitf^mi-sp#*/B6-*Mitf^Mi-b* and B6-*Mitf^mi-sp*/B6-*Mitf^Mi-b* animals.

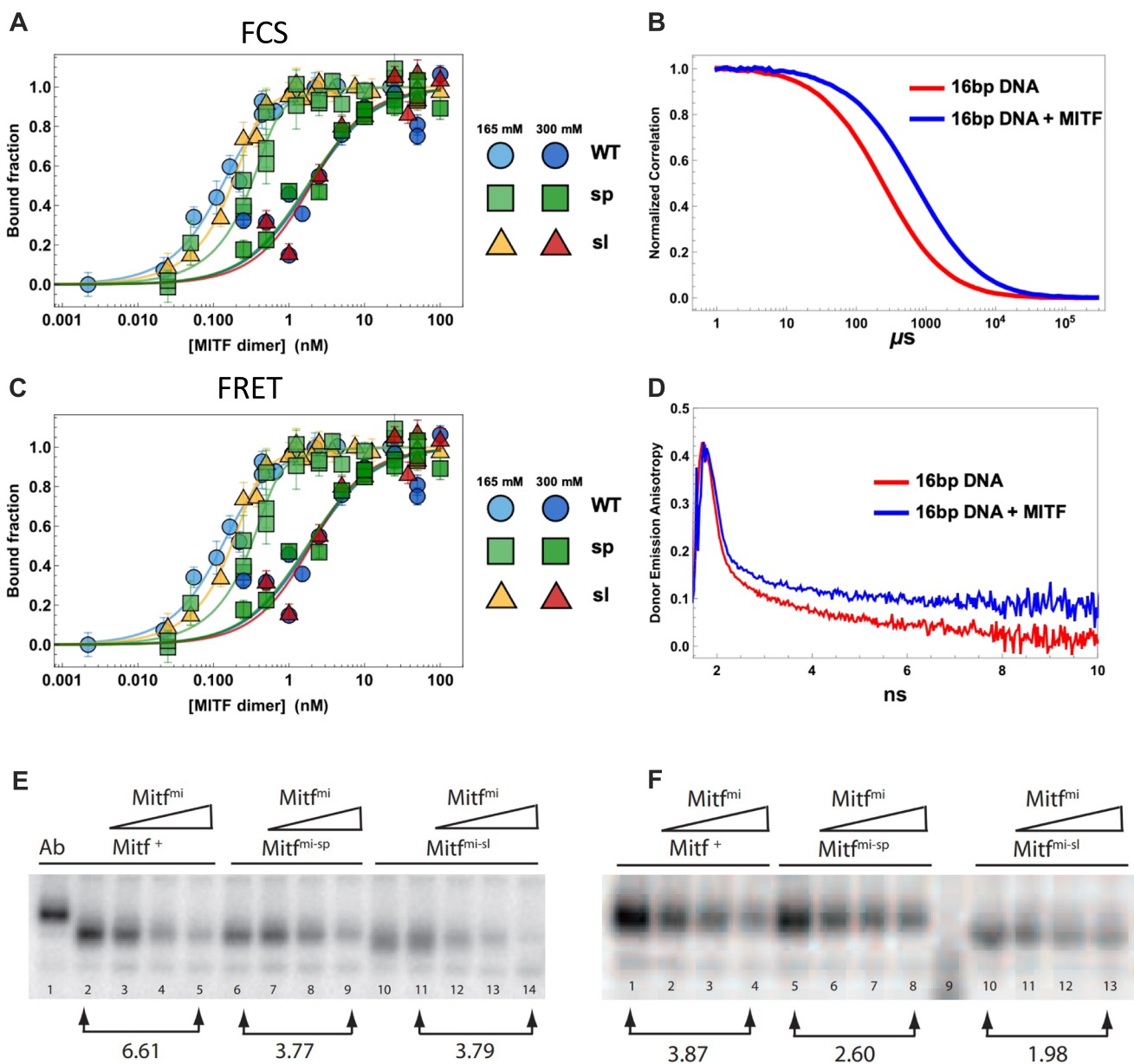

**Figure EV2.  The MITF-sl protein has similar DNA binding affinity, however, prefers to form dimers compared to MITF-WT and MITF-sp.**

(A) DNA binding curves of recombinantly expressed human MITF-WT, MITF-sp, and MITF-sl protein to M-box probe measured by Fluorescence Correlation Spectroscopy (FCS) at 165 mM KCl (blue) and 300 mM KCl (yellow). MITF-WT protein in circles, MITF-sp square boxes, and MITF-sl in triangles. Error bars represent two standard deviations of fit error at each point. (B) Normalized donor-acceptor fluorescence cross-correlation curves of 16bp M-Box DNA alone (red) and with 100 nM WT MITF added (blue), both at 300 mM KCl. (C) DNA binding curves of recombinantly expressed human MITF-WT, MITF-sp, and MITF-sl protein to M-box probe measured by mean FRET at 165 mM KCl (blue) and 300 mM KCl (yellow). MITF-WT protein in circles, MITF-sp square boxes, and MITF-sl in triangles. Error bars represent two standard deviations of fit error at each point. (D) Inverted time-correlated donor emission anisotropy of 16 bp DNA alone (red) and in the presence of 100 nM MITF WT (blue), both at 300 mM KCl. (E, F) Electrophoretic mobility shift assays were performed using the M-box sequence (5′-AAAGTCAGTCATGTGCTTTTCAGA-3′) as a probe. (E) MITF-WT, MITF-sp, and MITF-sl proteins were expressed using the TNT (Promega) system alone (lanes 1, 2, 6, 10, and 11) or co-expressed with the dominant-negative MITF-mi protein (lanes 3–5, 7–9, and 12–14) and then incubated with the labeled probe. The binding is specific since the presence of the C5 monoclonal MITF antibody which recognizes the N-terminus of Mitf results in a supershift (Ab). (F) The same experiment as in (E), except that the proteins were translated separately and then incubated for 30 min in the presence of DNA to allow heterodimerization before performing the mobility shift experiment.

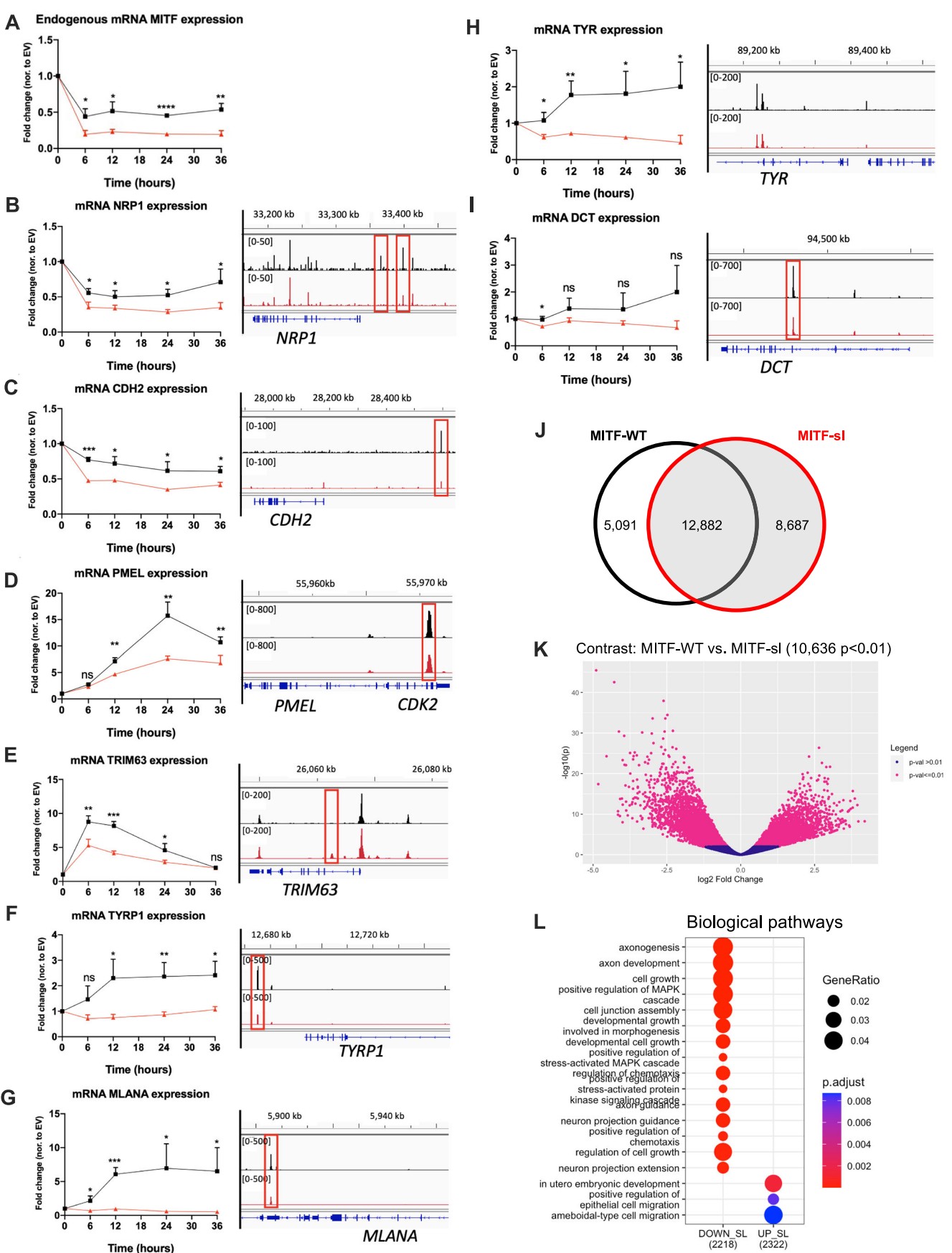

**Figure EV3.** **MITF-sl protein is a less potent activator than MITF-WT.**

RT-qPCR analysis and the CUT&RUN peaks in indicated genes from dox-inducible A375P cells overexpressing either MITF-WT or MITF-sl of (A) endogenous mRNA MITF and mRNA MITF target genes: (B) NRP1, (C) CDH2, (D) PMEL, (E) TRIM63, (F) TYRP1, (G) MLANA, (H) TYR, and (I) DCT in the dox-inducible A375P overexpressing cells. The cells were treated with doxycycline for 6, 12, 24, and 36 h to induce MITF expression at the same level before harvesting for RNA isolation. Actin and hAPR was used as housekeeping genes. The fold change in target gene expression was assessed in cells overexpressing either MITF-WT or MITF-sl by comparing to those expressing EV-FLAG-HA followed by normalization to the proportion of MITF proteins retained in the nucleus. Error bars represent SEM of at least three independent experiments. Statistically significant differences (Student's $t$ test) are indicated by *$P < 0.05$, **$P < 0.01$, ***$P < 0.001$, ****$P < 0.0001$, and ns not significant. (J) Venn Diagram showing the number of peaks shared and different between MITF-WT and the MITF-sl. (K) Peaks different between MITF-WT and MITF-sl mutant proteins shown in a Volcano plot ($P < 0.01$). (L) Gene ontology analysis of the 10,636 ($P < 0.01$) peaks that are different between MITF-WT and MITF-sl.

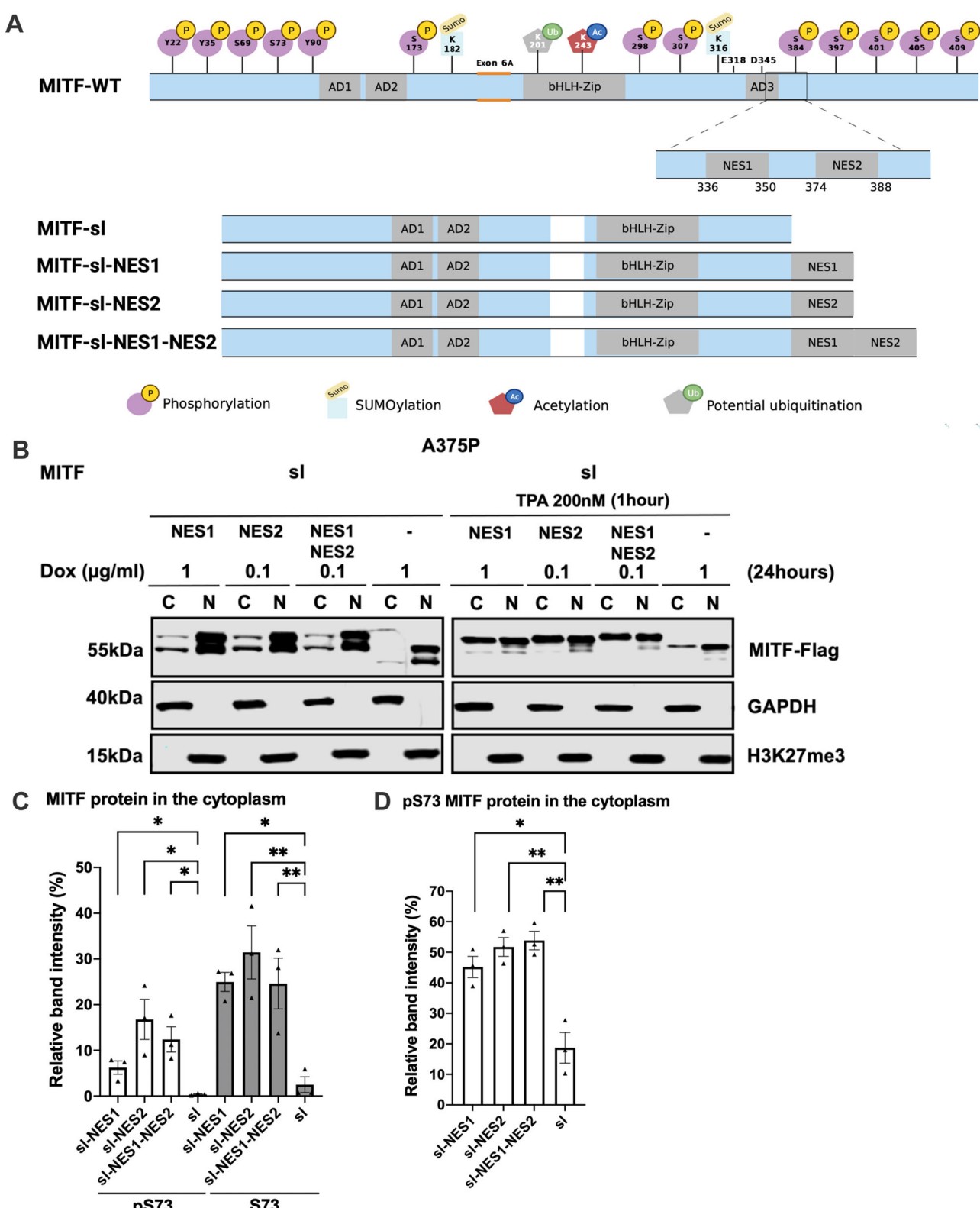

**Figure EV4. Identify two potential NES at the MITF C-terminus.**

(A) Graphical depiction of the MITF-WT, MITF-sl, MITF-sl-NES1, MITF-sl-NES2, and MITF-sl-NES1-NES2 proteins. The location of the NES1 and NES2 sequences in MITF-WT are also shown. (B) Western blot analysis of cytoplasmic (C) and nuclear (N) fractions from A375P melanoma cells induced for 24 h to overexpress the indicated MITF mutant proteins with or without treatment with 200 nM TPA for 1 h. MITF was visualized using FLAG antibody. GAPDH and H3K27me3 were loading controls for cytoplasmic and nuclear fractions, respectively. (C, D) MITF band intensities in the cytoplasmic and nuclear fractions from western blot analysis (B) were quantified separately with *ImageJ* software and are depicted as percentages of the total amount of protein present in the two fractions. Error bars represent SEM of three independent experiments. Statistically significant differences (Student's *t* test) are indicated by *$P < 0.05$ and **$P < 0.01$.

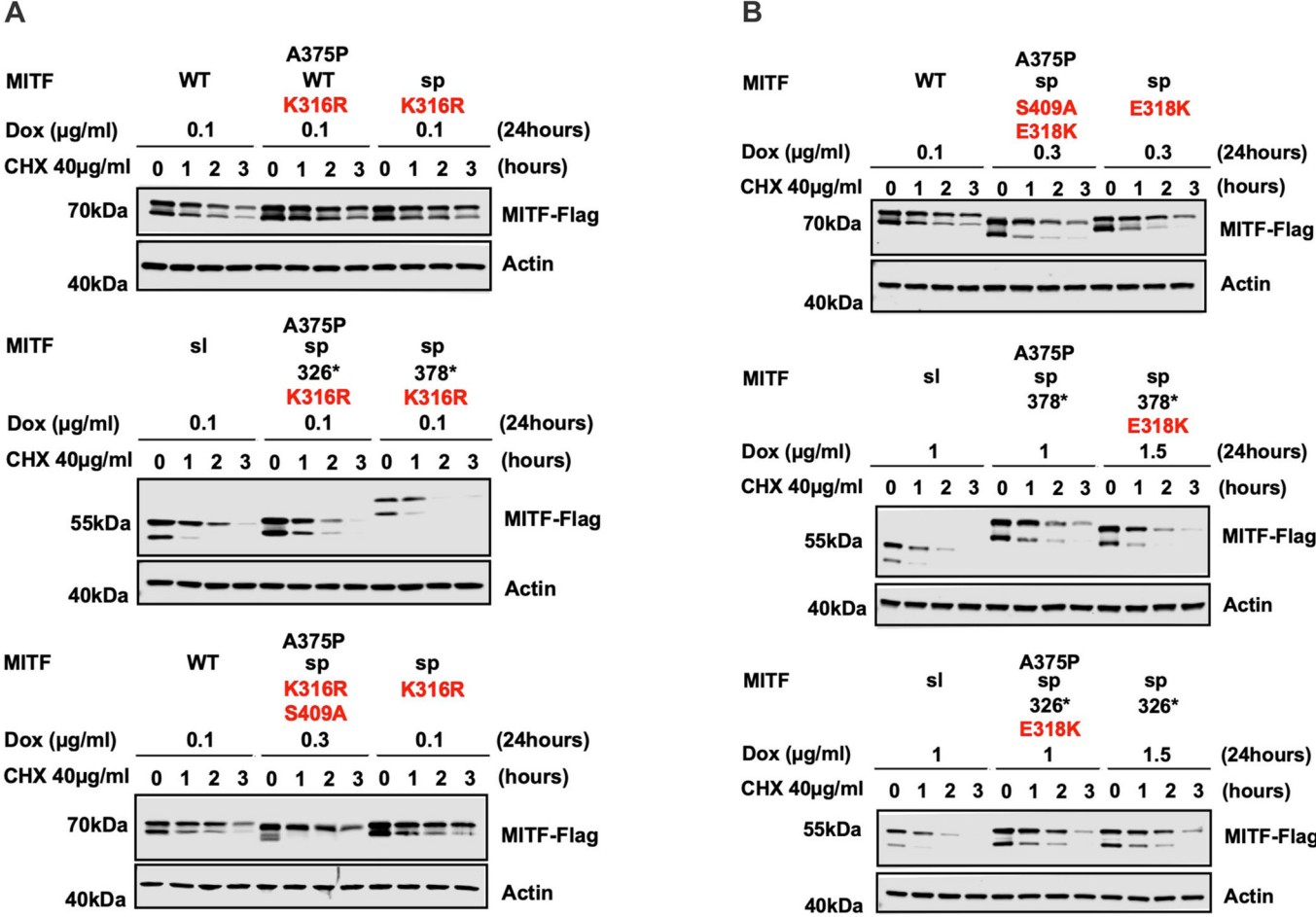

**Figure EV5.  K316R and E318K mutations effect stability of the MITF proteins.**

(A, B) Western blot analysis of the stability of the MITF proteins. The inducible A375P cells were treated with doxycycline for 24 h to express the indicated mutant MITF proteins before treating them with 40 μg/ml CHX for 0, 1, 2, and 3 h. The MITF protein was then compared by western blot using FLAG antibody. Actin was used as a loading control. The band intensities were quantified using ImageJ software.

