## [Peer Review File · EMBO Reports]

Novel mechanisms of MITF regulation identified in a mouse suppressor screen

Hong Nhung Vu, Matti Már Valdimarsson, Sara Sigurbjörnsdóttir, Kristín Bergsteinsdóttir, Julien Debbache, Keren Bismuth, Deborah A. Swing, Jón H. Halldsson, Lionel Larue, Heinz Arnheiter, Neal G. Copeland, Nancy A. Jenkins, Petur O. Heidarsson, and Eiríkur Steingrímsson

Corresponding author(s): Eiríkur Steingrímsson (eirikurs@hi.is)

Review Timeline:

Submission Date:	22nd Feb 24
Editorial Decision:	12th Apr 24
Revision Received:	21st May 24
Editorial Decision:	2nd Jul 24
Revision Received:	8th Jul 24
Accepted:	17th Jul 24

Transaction Report:

Dear Prof. Steingrímsson

Thank you for the submission of your research manuscript to our journal. I apologize for the delay in handling your manuscript, but we have now received the full set of referee reports that is copied below.

As you will see, the referees acknowledge that the findings are potentially interesting and that the conclusions are overall supported by the data presented but they also raise a number of concerns and have suggestions how to further strengthen the data. Referee #2 and #3 both comment on data presentation and I agree with their concerns regarding the text and the suggestion to shorten it to present the data and conclusions in a more concise way.

Given these constructive comments, we would like to invite you to revise your manuscript with the understanding that the referee concerns (as detailed above and in their reports) must be fully addressed and their suggestions taken on board. Referee #1 suggests testing the effect of the MITF suppressor mutation on melanoma, which can likely be addressed in a cell culture model of melanoma cells, I suppose.

Please address all referee concerns in a complete point-by-point response. Acceptance of the manuscript will depend on a positive outcome of a second round of review. It is EMBO Reports policy to allow a single round of revision only and acceptance or rejection of the manuscript will therefore depend on the completeness of your responses included in the next, final version of the manuscript.

We realize that it is difficult to revise to a specific deadline. In the interest of protecting the conceptual advance provided by the work, we recommend a revision within 3 months (July 12). Please discuss the revision progress ahead of this time with the editor if you require more time to complete the revisions.

I am also happy to discuss the revision further via e-mail or a video call, if you wish.

Please note that all panels of a figure must fit on one page. Currently you supplied several pages for one figure. Some of the figures might be split in two if this enhances presentation and readability of the data. We have no limitation for the number of figures in Articles.

*****IMPORTANT NOTE:

We perform an initial quality control of all revised manuscripts before re-review. Your manuscript will FAIL this control and the handling will be delayed IN CASE the following APPLIES:

- 1) A data availability section providing access to data deposited in public databases is missing. If you have not deposited any data, please add a sentence to the data availability section that explains that.
- 2) Your manuscript contains statistics and error bars based on $n=2$. Please use scatter blots in these cases. No statistics should be calculated if $n=2$.

When submitting your revised manuscript, please carefully review the instructions that follow below. Failure to include requested items will delay the evaluation of your revision.*****

2) individual production quality figure files as .eps, .tif, .jpg (one file per figure).

Please download our Figure Preparation Guidelines (figure preparation pdf) from our Author Guidelines pages <https://www.embopress.org/page/journal/14693178/authorguide> for more info on how to prepare your figures.

4) a complete author checklist, which you can download from our author guidelines

(<<https://www.embopress.org/page/journal/14693178/authorguide>>). Please insert information in the checklist that is also reflected in the manuscript. The completed author checklist will also be part of the RPF.

5) Please note that all corresponding authors are required to supply an ORCID ID for their name upon submission of a revised manuscript (<<https://orcid.org/>>). Please find instructions on how to link your ORCID ID to your account in our manuscript tracking system in our Author guidelines

(<<https://www.embopress.org/page/journal/14693178/authorguide#authorshipguidelines>>)

6) We replaced Supplementary Information with Expanded View (EV) Figures and Tables that are collapsible/expandable online. A maximum of 5 EV Figures can be typeset. EV Figures should be cited as 'Figure EV1, Figure EV2' etc... in the text and their respective legends should be included in the main text after the legends of regular figures.

7) Please note that a Data Availability section at the end of Materials and Methods is now mandatory. In case you have no data that requires deposition in a public database, please state so instead of refereeing to the database.

See also < <https://www.embopress.org/page/journal/14693178/authorguide#dataavailability>>. Please note that the Data Availability Section is restricted to new primary data that are part of this study.

Additional information on source data and instruction on how to label the files are available

<<https://www.embopress.org/page/journal/14693178/authorguide#sourcedata>>.

10) Figure legends and data quantification:

- the name of the statistical test used to generate error bars and P values,
- the number (n) of independent experiments (please specify technical or biological replicates) underlying each data point,
- the nature of the bars and error bars (s.d., s.e.m.)

- If the data are obtained from n {less than or equal to} 5, show the individual data points in addition to the SD or SEM.

- If the data are obtained from n {less than or equal to} 2, use scatter blots showing the individual data points.

11) Our journal encourages inclusion of *data citations in the reference list* to directly cite datasets that were re-used and obtained from public databases. Data citations in the article text are distinct from normal bibliographical citations and should directly link to the database records from which the data can be accessed. In the main text, data citations are formatted as follows: "Data ref: Smith et al, 2001" or "Data ref: NCBI Sequence Read Archive PRJNA342805, 2017". In the Reference list, data citations must be labeled with "[DATASET]". A data reference must provide the database name, accession number/identifiers and a resolvable link to the landing page from which the data can be accessed at the end of the reference. Further instructions are available at <<https://www.embopress.org/page/journal/14693178/authorguide#referencesformat>>.

12) All Materials and Methods need to be described in the main text. We would encourage you to use 'Structured Methods', our new Methods format. According to this format, the Methods section should include a Reagents and Tools Table (listing key reagents, experimental models, software and relevant equipment and including their sources and relevant identifiers) followed by a Methods and Protocols section in which we encourage the authors to describe their methods using a step-by-step protocol

format with bullet points, to facilitate the adoption of the methodologies across labs. More information on how to adhere to this format as well as downloadable templates (.doc or .xls) for the Reagents and Tools Table can be found in our author guidelines: < <https://www.embopress.org/page/journal/14693178/authorguide#manuscriptpreparation>>.

An example of a Method paper with Structured Methods can be found here:
<<https://www.embopress.org/doi/10.15252/msb.20178071>>.

13) As part of the EMBO publication's Transparent Editorial Process, EMBO Reports publishes online a Review Process File to accompany accepted manuscripts. This File will be published in conjunction with your paper and will include the referee reports, your point-by-point response and all pertinent correspondence relating to the manuscript.

Kind regards,

Referee #1:

In this study, Vu and coauthors discovered a novel intragenic MITF mutation identified through a MITF pigmentation suppressor mouse screen. The resulting truncated MITF protein effected both an increase in nuclear localization and decreased stability compared to WT MITF. The truncated protein also dimerizes to WT and mutant MITF which helps mediate the localization and stabilization effects of the normalized MITF.

As MITF is the master regulator of melanocyte development and a key gene present in melanomagenesis, identification of novel mutations is important in progressing the field. The authors do an extremely thorough job in exploring the structural and colocalization consequences of expressing this mutation. While the authors looked at proliferation assays and performed RNA-Seq, performing a few more phenotypic assays would be helpful in strengthening the manuscript.

As MITF and its associated genes are important for phenotype switching in melanoma, how does the MITF pigmentation suppressor mutation effect melanoma migration?

How does this mutation mediate cellular stress effects on melanoma cells, ie UV irradiation and BRAF/MEK inhibition?

Referee #2:

Vu HN, et al. present a very thorough and well executed study that not only provides novel information regarding a transcription factor of great importance for pigment cell field but also reports a creative methodology for identifying novel genetic suppressor mechanism that will be of interest to a broader molecular biology audience.

The work has several strengths: 1) Vu HN, et al. utilize the rich history and repertoire of pigment associated genotype-phenotype mouse lines to design an eloquent method for their discovery-based approach. 2) This led to the discovery of new genetic suppressor mechanism; while also 3) identifying mechanisms of MITF stability, subcellular location, and transcriptional activity important for melanocyte homeostasis, adaptation, and disease progression.

However, there are several major and minor comments that should be addressed before the manuscript is considered for publication.

Major comments:

1) The article, both in text and figures, appears excessively lengthy. Although it falls within the character limit, it reads as though

it were twice as long. The manuscript could benefit from a more concise description of findings and the exclusion of unsubstantiated statements (i.e. statements with no references or supporting data) that are not important for the narrative. The reviewer also suggests exploring alternative ways to present data, facilitating comparison between conditions and mutants with fewer figure panels, thereby potentially reducing the number of supplemental figures.

2) Regarding Figure 1J, it is customary to include the electropherogram. Additionally, the reviewer requests clarification on the methods used for DNA sequencing that confirmed the Sanger sequencing.

3) In lines 191-192, the authors mention that A375P melanoma cells express little endogenous MITF. The reviewer seeks clarification on the basis for this statement. Have the authors compared expression levels to other melanoma cell lines, or are there previous studies supporting this claim? References or data should be provided to substantiate this statement.

4) In lines 199-201, the authors suggest there are generally fewer melanocytes in embryos with the two mutants. However, the quantification in Figure S2D suggests this is only true for one of the three locations/time points (E12.5 eye region). Moreover, the quantification does not seem consistent with the representative images in S2C. The reviewer points out that if anything, it appears as if the two mutants yield more melanocytes at the later E15.5 time points in all locations. The written interpretation provided by the authors does not seem to align with the majority of the data in S2C,D.

5) Regarding the in vitro differentiation of neural crest cells, the reviewer questions whether it is expected that only melanocytes will express MITF. If so, the authors should provide supporting evidence or citations. Additionally, the methods for immunofluorescence are missing and should be included. The reviewer also raises concerns about the specificity of the C5 anti-MITF antibody and requests evidence that other lineages derived during in-vitro differentiation of NCCs do not express MITF isoforms recognized by C5. It is possible that the authors will need to co-stain with other melanocytic and non-melanocyte lineage markers that indicate the MITF positive cells are committed to the melanocyte fate.

6) Also concerning the in vitro differentiation of neural crest cells: How were pigmented cells counted? Was it normalized to total number of cells? It is unclear how pigmented cells were counted as it is difficult to detect individual cells based on the representative images and there is limited information in the methods.

7) The authors claim in lines 214-215 that the partial function of the mutant MITF protein is "not caused by prolonged cell proliferation, affecting differentiation". The reviewer asks if there is data supporting this statement.

8) For the mouse MITF-M overexpression in A375P melanoma cells, the reviewer asks whether it is known that mouse MITF-M can rescue human MITF-M. This is important to highlight in the main text since several assays include mouse MITF expression in human cells, including evaluation of transcriptional changes.

9) The authors conclude that MITF-sl may have stronger dimerization affinity than MITF-WT (lines 227-228). The reviewer recommends providing quantification of hetero- vs homo-dimer in Figure S4 to further support this statement and other more definitive statements based on the figure.

10) The authors switch to human MITF-WT, -sp, and -sl proteins for the DNA affinity assay. The reviewer asks if this is the M isoform and suggests discussing the homology of human and mouse MITF-M isoforms and whether this impacts the location of -sp and -sl mutations in the human gene/protein.

11) Regarding Figures 1L and S5A&C, the reviewer suggests making it clearer which lines belong to which MITF protein. They recommend changing the size of the symbols and altering the appearance of the lines for each KCl condition. Additionally, the reviewer asks if there are statistics to support potential differences in the binding affinities for the various methods.

12) The reviewer seeks an explanation for why only one MITF band is present for most samples in Figure 5D.

13) It appears Figure panels 6A and 6B are switched. The reviewer notes that the main text refers to intramolecular FRET first and references Figure 6A, but Figure 6A (and legend) is intermolecular FRET. Additionally, the reviewer asks about the grey peaks with grey shaded boxes in 6A-C. This information should be provided in the figure legend or main text.

14) The authors should provide at minimum the list, and accompanying statistics, for the differential expression analysis from their RNAseq study. Ideally, they should also deposit the sequencing data into a publicly available database.

15) In the discussion, the authors state that their finding of the S73 form of MITF-WT is much less stable than the pS73 form contrasts with previous literature. The reviewer suggests further discussion on why the data are not consistent and providing an explanation for this inconsistency.

Minor Comments:

- 1) There may be a typo in line 134 regarding which mouse is darkly pigmented in Figure S1B. The text states "right" mouse but it appears to be the mouse on the left.
- 2) Figure 1E is mentioned out of order in the text. The authors should either revise the text or revise the order of figure panels so each panel is referenced in the main text in the order they appear.
- 3) A summary schematic would be helpful in summarizing the findings and the model the authors propose at the end of the discussion.

Referee #3:

The work of Vu et al. extensively characterizes a mutation the group induced in MITF, and characterizing its effect on protein stability, DNA binding, gene expression, and others. The work is expansive, and the authors applied diverse methods to gain new insights into the regulation and function of MITF.

The arrangement of the story makes it a little hard to follow. For instance, many seemingly key findings including ChIP-seq and RNA-seq being touched upon in the supplement rather than in the main manuscript. Whereas more minor findings not relevant for the main story are expanded on in the text and presented in main figures.

129 - How does the darker coat "the presence of a mutagenized chromosome", and what does mutagenized chromosome mean?

154 - "Similarly, no effects were observed on eye size or bone development" where was this quantified or shown?

256 - "Similar observations were made when RNA obtained from the skin was quantitated (Figure S6)". It doesn't seem like similar results were actually found? Further, it is questionable whether quantifying MITF this way will yield meaningful results, as the number of MITF expressing cells may vary.

359 - "a significant portion of the MITF-sp-326* and MITF-sp-378* proteins was present in the nuclear fraction". The authors do not provide statistical comparison for this.

Figure S7. It is unclear how this normalization was conducted. "EV-FLAG-HA cell lines and then to the proportion of MITF proteins retained in the nucleus." Was a housekeeping gene used? Authors should provide quantitated results for their claim that the proteins were expressed at equal level.

Figure 2. I,J,K Was the data normalized to the actin control?

Dr. Martina Rembold, PhD
Senior Editor
EMBO reports

21.5.2024

Dear Dr. Rembold,

We thank you and the reviewers for the valuable feedback on our manuscript. As suggested, we have shortened the manuscript by removing non-essential sections, including the analysis of pigmentation and appearance of pigment cells in the mutants. These experiments did not contribute significantly to the understanding of the suppressor mutation. We have enhanced the readability of the figures which now all fit on one page. The changes to the figures are explained in the table below. We have formatted the manuscript according to instructions (we did not track the changes to the references in the tracked-version of the manuscript). Source data have been dealt with according to instructions.

We are as interested as the reviewers in understanding how the E318K variant and the C-end of MITF are involved in cancer predisposition. We have performed numerous experiments to characterize the role of the MITF-sl in melanoma and have, for example, determined what happens upon treatment with the BRAF-inhibitor Vemurafenib (see details in the replies to the reviewers below). However, each experiment has added a layer of complexity, and our results are far from clarifying how E318K predisposes to melanoma or if the C-end of MITF contributes to the process. We think that a lot more experiments are needed, including extensive animal models, before we can make claims about this. Therefore, we decided to leave this discussion out of the current manuscript.

Below, you will find replies to each of the reviewers' comments. We believe we have addressed all the issues raised and hope the manuscript is now suitable for publication in EMBO Reports.

Best regards,

Eiríkur Steingrímsson
Department of Biochemistry and Molecular Biology
BioMedical Center
Faculty of Medicine
University of Iceland
eirikurs@hi.is

Replies to reviewers

We thank the reviewers for their valuable feedback on our manuscript. As suggested, we have shortened the manuscript by removing non-essential sections, including the analysis of pigmentation and appearance of pigment cells in the mutants. These experiments did not contribute significantly to the understanding of the suppressor mutation or were replaced by better methods. We have also enhanced the readability of the figures which now all fit on one page. The changes to the figures are explained in the table below.

Below we address each of the comments raised.

Replies to individual comments.

Referee #1:

In this study, Vu and coauthors discovered a novel intragenic MITF mutation identified through a MITF pigmentation suppressor mouse screen. The resulting truncated MITF protein effected both an increase in nuclear localization and decreased stability compared to WT MITF. The truncated protein also dimerizes to WT and mutant MITF which helps mediate the localization and stabilization effects of the normalized MITF.

As MITF is the master regulator of melanocyte development and a key gene present in melanomagenesis, identification of novel mutations is important in progressing the field. The authors do an extremely thorough job in exploring the structural and colocalization consequences of expressing this mutation. While the authors looked at proliferation assays and performed RNA-Seq, performing a few more phenotypic assays would be helpful in strengthening the manuscript. As MITF and its associated genes are important for phenotype switching in melanoma, how does the MITF pigmentation suppressor mutation effect melanoma migration?

How does this mutation mediate cellular stress effects on melanoma cells, ie UV irradiation and BRAF/MEK inhibition?

Reply: We thank the reviewer for his positive comments.

We analyzed the proliferation of the human embryonic kidney cell line HEK293 and the A375P melanoma cells overexpressing WT, MITF-sp, and MITF-sl proteins to determine if the suppressor mutation increased pigmentation by increasing the number of melanocytes. We did not see a difference between these isoforms and now this section of the manuscript has been removed. Although we have not tested if the suppressor mutation affects migration, our analysis of melanoblasts during embryogenesis does not indicate differences between WT, MITF-sp, and MITF-sl in this respect. We have tested if BRAF/MEK inhibition affects protein stability and saw that the stability of the S73 phosphorylated (pS73) MITF-WT and MITF-sp proteins was significantly reduced upon PLX-4032 treatment, whereas the stability of the unphosphorylated (S73) MITF-WT and MITF-sp was not changed (see Figure A below). However, the stability of the MITF-sl protein was not altered upon PLX-4032 treatment, regardless of phosphorylation status (Figure A). Currently, we do not understand what this difference means in terms of function, if anything. We have performed several experiments to investigate the role of the E318K variant and the MITF-sl mutation in melanoma, including Cut-n-Run, RT-qPCR, and proteomics (mass spec) experiments (not included in the current manuscript). However, each experiment has added a layer of complexity to the story, and so far our results have not explained how E318K affects melanoma predisposition. We think that a lot of further experiments are needed to dissect what E318K variant does, how it is different from K316R, and whether the MITF-sl mutation contributes to melanoma, including mouse models. The focus of this manuscript is on the molecular nature of the MITF-sl suppressor

mutation. As the reviewer notes, the story is already complicated. We therefore think that adding a role in melanoma is premature at this point. We have therefore de-emphasized the melanoma link e.g. by removing the analysis on effects of proliferation.

Figure for referee with unpublished data and its description has been removed upon request by the authors.

Referee #2:

Vu HN, et al. present a very thorough and well executed study that not only provides novel information regarding a transcription factor of great importance for pigment cell field but also reports a creative methodology for identifying novel genetic suppressor mechanism that will be of interest to a broader molecular biology audience. The work has several strengths: 1) Vu HN, et al. utilize the rich history and repertoire of pigment associated genotype-phenotype mouse lines to design an eloquent method for their discovery-based approach. 2) This led to the discovery of new genetic suppressor mechanism; while also 3) identifying mechanisms of MITF stability, subcellular location, and transcriptional activity important for melanocyte homeostasis, adaptation, and disease progression.

Reply: We thank the reviewer for the positive comments.

However, there are several major and minor comments that should be addressed before the manuscript is considered for publication.

Major comments:

1) The article, both in text and figures, appears excessively lengthy. Although it falls within the character limit, it reads as though it were twice as long. The manuscript could benefit from a more concise description of findings and the exclusion of unsubstantiated statements (i.e. statements with no references or supporting data) that are not important for the narrative. The reviewer also suggests exploring alternative ways to present data, facilitating comparison between conditions and mutants with fewer figure panels, thereby potentially reducing the number of supplemental figures.

Reply: We have shortened the text extensively and tried to present the data more simply. For example, to shorten the manuscript we have removed supplementary figures S2A-D and S3 and all the text referring to these figures (see points 4-7 below). We also simplified the Figures and graphs as indicated in the table below that tracks the changes we made to the Figures).

2) Regarding Figure 1J, it is customary to include the electropherogram. Additionally, the reviewer requests clarification on the methods used for DNA sequencing that confirmed the Sanger sequencing.

Reply: We have added the following to the section on qRT-PCR and sequencing in the Materials and Methods chapter: "The results were confirmed by sequencing additional animals as well as several controls animals, on which the mutation was induced, in order to confirm the alteration". This is also emphasized in the results section.

3) In lines 191-192, the authors mention that A375P melanoma cells express little endogenous MITF. The reviewer seeks clarification on the basis for this statement. Have the authors compared expression levels to other melanoma cell lines, or are there previous studies supporting this claim? References or data should be provided to substantiate this statement.

Reply: The low expression of MITF in A375P cells is described in Wouters et al. 2020 (PMID: 32753671). We have added this reference to the manuscript in the appropriate place.

4) In lines 199-201, the authors suggest there are generally fewer melanocytes in embryos with the two mutants. However, the quantification in Figure S2D suggests this is only true for one of the three locations/time points (E12.5 eye region). Moreover, the quantification does not seem consistent with the representative images in S2C. The reviewer points out that if anything, it appears as if the two mutants yield more melanocytes at the later E15.5 time points in all locations. The written interpretation provided by the authors does not seem to align with the majority of the data in S2C,D.

Reply: see after comment 7.

5) Regarding the in vitro differentiation of neural crest cells, the reviewer questions whether it is expected that only melanocytes will express MITF. If so, the authors should provide supporting evidence or citations. Additionally, the methods for immunofluorescence are missing and should be included. The reviewer also raises concerns about the specificity of the C5 anti-MITF antibody and requests evidence that other lineages derived during in-vitro differentiation of NCCs do not express MITF isoforms recognized by C5. It is possible that the authors will need to co-stain with other

melanocytic and non-melanocyte lineage markers that indicate the MITF positive cells are committed to the melanocyte fate.

Reply: see after comment 7.

6) Also concerning the in vitro differentiation of neural crest cells: How were pigmented cells counted? Was it normalized to total number of cells? It is unclear how pigmented cells were counted as it is difficult to detect individual cells based on the representative images and there is limited information in the methods.

Reply: see after comment 7.

7) The authors claim in lines 214-215 that the partial function of the mutant MITF protein is "not caused by prolonged cell proliferation, affecting differentiation". The reviewer asks if there is data supporting this statement.

Reply to 4-7: After careful consideration of the reviewer's comments, we have decided to remove the discussion mentioned in reviewer points 4-7 from the manuscript. These experiments (described in Figures S2A-D and S3 and all the text referring to these Figures) were performed during our journey to understand how MITF-sl mediates phenotypic suppression and, at the time, helped us eliminate certain possibilities and to focus our work. However, we think these results are not necessary at this point. First, the experiments described in this section were performed on the wt, MITF-sp, and MITF-sl mutations in isolation and not when they were in the presence of the other mutations that MITF-sl suppresses. Second, the data are mostly negative and report no important differences and, therefore, do not explain the suppression phenotype or how the mutation affects function. Third, the reviewers ask for simplification of the manuscript and we think that removing this section is a helpful step in that direction.

For completeness sake, following are replies to each of items 4-7:

4) Figures S2 C, D measures Kit-positive cells in vivo (from a transgene), while Fig S3A (referred to by lines 199-201) measures melanin-positive cells in explant cultures. These assays are not necessarily the same as far as dynamics are concerned.

5) The C5 antibody recognizes an epitope in MITF that is located between residues 120 and 170 of MITF (Fock et al., 2018, PMID: 29938923), a domain that is not highly conserved between the MITF, TFEB, and TFEC proteins. Although we have not tested whether C5 may crossreact with TFEB and TFE3, these two genes are expressed at lower levels in melanoma cells and we do not see these proteins on western blots from melanocytes or melanoma cells (Fock et al., 2018). However, the presence of pigmentation indicates that these cells are positive for MITF since MITF is the master regulator of melanocytes. We, therefore, do not think it is likely that the other family members are visible in our assays.

6) They were counted over the entire culture per field of vision. They were not normalised to total cell numbers. Even though the cultures were of various sizes, there was no systematic size difference between sl and wt cultures. At high resolution, melanin-positive cells can be counted.

7) We have not systematically looked at proliferation by BrdU incorporation in neural crest cell cultures or in vivo, so this statement has been deleted.

8) For the mouse MITF-M overexpression in A375P melanoma cells, the reviewer asks whether it is known that mouse MITF-M can rescue human MITF-M. This is important to highlight in the main text since several assays include mouse MITF expression in human cells, including evaluation of transcriptional changes.

Reply: We used the mouse cDNA as backbone for all our studies, except the direct DNA-binding and structural studies. We wanted to be able to distinguish the exogenous gene from the endogenous one when transfecting into human cells. To our knowledge, rescue experiments have not been performed where the human MITF-M gene has been used to rescue mouse MITF-M function. The mouse and human MITF-M proteins share 94% sequence identity and most of the changes are conservative. Numerous experiments have been performed using the mouse and human MITF-M proteins and so far no differences have been found in function, effects on gene activation or expression. To address this concern we have added an explanation in the results chapter that we used the mouse MITF-M protein in human cells in all experiments except the DNA-binding and structural studies.

9) The authors conclude that MITF-sl may have stronger dimerization affinity than MITF-WT (lines 227-228). The reviewer recommends providing quantification of hetero- vs homo-dimer in Figure S4 to further support this statement and other more definitive statements based on the figure.

Reply: Figure S4 (Figure S2 in revised version) shows fluorescent images of SDS page gels that are the second dimension after running blue-native PAGE. We are hesitant to quantitate these bands as they are derived from 2D electrophoresis and are large and diffuse and thus it is difficult to set the borders. We have, therefore, removed the statement.

10) The authors switch to human MITF-WT, -sp, and -sl proteins for the DNA affinity assay. The reviewer asks if this is the M isoform and suggests discussing the homology of human and mouse MITF-M isoforms and whether this impacts the location of -sp and -sl mutations in the human gene/protein.

Reply: The structural experiments were indeed performed using human MITF-M. The difference between the human and mouse MITF-M proteins is minimal (see reply to 8 here above). This difference does not impact the location of the sp and sl mutations as both positions are fully conserved. This is now clearly explained in the text.

11) Regarding Figures 1L and S5A&C, the reviewer suggests making it clearer which lines belong to which MITF protein. They recommend changing the size of the symbols and altering the appearance of the lines for each KCl condition. Additionally, the reviewer asks if there are statistics to support potential differences in the binding affinities for the various methods.

Reply: The figure has been changed according to the reviewer's suggestions. Our conclusion of negligible differences in binding affinities between WT and variant MITF proteins is based on measurements on a very large number of individual molecules based on three independent spectroscopic variables that gave remarkably similar results. The FRET data is based on ratiometric measurements of donor and acceptor photons, anisotropy data is based on the ratio of vertical to horizontally polarized emission photons, and the FCS data is based on statistics of photon timings. Each data point in the binding isotherms corresponds to an average of >5000 molecules, which we have now described in the figure legend, and the total number of individual measurements of FRET and anisotropy exceeds 50000 molecules. We report fit errors which are useful to assess the quality of the data and the overlap of standard deviations strongly support similar binding affinities. Finally, the consistency of the experimental outcomes was verified through repeat trials on freshly prepared samples using two different experimental

conditions, i.e. at different ionic strengths (not shown). We are therefore confident that our experiments yield accurate measures of binding affinities and that our main conclusion holds, that any difference between WT and variants is negligible.

12) The reviewer seeks an explanation for why only one MITF band is present for most samples in Figure 5D.

Reply: In 5D (now 5C), all the samples were treated with TPA for one hour. TPA treatment of MITF leads to phosphorylation of S73 in all of MITF proteins present such that none of the lower non-phosphorylated protein is detected. This also moves most of the protein to the cytoplasm. We have added this explanation to the figure legend.

13) It appears Figure panels 6A and 6B are switched. The reviewer notes that the main text refers to intramolecular FRET first and references Figure 6A, but Figure 6A (and legend) is intermolecular FRET. Additionally, the reviewer asks about the grey peaks with grey shaded boxes in 6A-C. This information should be provided in the figure legend or main text.

Reply: The switched labels in the figures have been fixed. An explanation of the grey peaks has been added to the figure legend.

14) The authors should provide at minimum the list, and accompanying statistics, for the differential expression analysis from their RNAseq study. Ideally, they should also deposit the sequencing data into a publicly available database.

Reply: We have not performed an RNAseq study but used qPCR to analyze gene expression. All the necessary statistics were performed for the qPCR studies, as presented in Figure S7 (Figure S4 in revised version). We did, however, perform a Cut-n-Run analysis for DNA binding, and this data has been deposited into GEO database under accession number GSE267956. We also add the list of genes exhibiting statistically significant differences ($p < 0.01$) between MITF-WT and MITF-sl as a supplemental table 3

15) In the discussion, the authors state that their finding of the S73 form of MITF-WT is much less stable than the pS73 form contrasts with previous literature. The reviewer suggests further discussion on why the data are not consistent and providing an explanation for this inconsistency.

Reply: Our results differ from that of Wu et al. (2000, PMID: 10673502). We expressed the mouse MITF-M protein in human melanoma cells (primarily A375P but also SKmel28 and 501mel cells) and then determined the stability of the phosphorylated and unphosphorylated proteins by tracing the tagged wild type and mutant MITF-proteins. Wu et al. (2000) treated the cells with recombinant Steel factor (MGF) and a MEK inhibitor and observed that the phosphorylated protein was degraded faster than the unphosphorylated protein. They also used TPA to stimulate phosphorylation and generated a S73A mutant to prevent phosphorylation and verified their results. At this point we have no other explanation for this difference than the systems used. It should be noted, however, that the dynamic nature of the process adds a layer of complication where the protein is continuously being phosphorylated and de-phosphorylated and the protein moved between compartments based on phosphorylation (Ngeow, 2018, PMID: 30150413). Unfortunately, we know very little about the dynamics of the phosphorylation and nuclear export processes. We have added a sentence to the discussion explaining this.

Minor Comments:

1) There may be a typo in line 134 regarding which mouse is darkly pigmented in Figure S1B. The text states "right" mouse but it appears to be the mouse on the left

Reply: The text has been revised and fixed.

2) Figure 1E is mentioned out of order in the text. The authors should either revise the text or revise the order of figure panels so each panel is referenced in the main text in the order they appear.

Reply: The text has been revised such that the order of figures is appropriate.

3) A summary schematic would be helpful in summarizing the findings and the model the authors propose at the end of the discussion.

Reply: We have included a graphical abstract (see below) that we hope appropriately summarizes the discussion.

Referee #3:

The work of Vu et al. extensively characterizes a mutation the group induced in MITF, and characterizing its effect on protein stability, DNA binding, gene expression, and others. The work is expansive, and the authors applied diverse methods to gain new insights into the regulation and function of MITF.

The arrangement of the story makes it a little hard to follow. For instance, many seemingly key findings including ChIP-seq and RNA-seq being touched upon in the supplement rather than in the main manuscript. Whereas more minor findings not relevant for the main story are expanded on in the text and presented in main figures.

Reply: We thank the reviewer for positive comments. We have shortened the discussion extensively and removed some of the more minor findings. We have also tried to simplify the figures as described in the table below. The ChIP-seq studies showed that the MITF-sl protein binds to a lot more sites in the genome than the wt protein and is in that sense similar to MITF-

E318K. However, this analysis did not add further depth to our analysis and therefore we placed it in the supplemental figures. We did not present RNAseq data in this manuscript.

129 - How does the darker coat "the presence of a mutagenized chromosome", and what does mutagenized chromosome mean?

Reply: We have changed the phrasing to "the presence of a suppressor mutation" to avoid confusion.

154 - "Similarly, no effects were observed on eye size or bone development" where was this quantified or shown?

Reply: The reviewer is correct that we did not quantitate eye size or bone development in the mice. Thus we have changed the phrasing to "No obvious changes were observed in eye size or bone development".

256 - "Similar observations were made when RNA obtained from the skin was quantitated (Figure S6)". It doesn't seem like similar results were actually found? Further, it is questionable whether quantifying MITF this way will yield meaningful results, as the number of MITF expressing cells may vary.

Reply: The reviewer is correct and we have removed this statement.

359 - "a significant portion of the MITF-sp-326* and MITF-sp-378* proteins was present in the nuclear fraction". The authors do not provide statistical comparison for this.

Reply: Statistical comparison for this can be found in Figure 3C

Figure S7. It is unclear how this normalization was conducted. "EV-FLAG-HA cell lines and then to the proportion of MITF proteins retained in the nucleus." Was a housekeeping gene used? Authors should provide quantitated results for their claim that the proteins were expressed at equal level.

Reply: We employed Actin and hARP as housekeeping genes. The fold change in gene expression was determined by comparing cells overexpressing MITF-WT or MITF-sl to those overexpressing EV-FLAG-HA. To ensure comparable protein levels of MITF-WT and MITF-sl overexpression, we treated them with varying dox concentrations to achieve approximately the same protein concentration, as indicated in all figures where western blot analyses is presented. Despite achieving similar protein levels, variations in the cytoplasmic-to-nuclear ratio required us to continuously normalize fold change with the proportion of MITF protein retained in the nucleus, ensuring the most accurate and reliable results possible. To make it more clear, we changed the phrasing to "Actin and hAPR were used as housekeeping genes. The fold change in target gene expression was assessed in cells overexpressing either MITF-WT or MITF-sl by comparing to those expressing EV-FLAG-HA followed by normalization to the proportion of MITF proteins retained in the nucleus". We hope this clarifies the issue.

Figure 2. I,J,K Was the data normalized to the actin control?

Reply: Yes the data was normalized to actin control. This has been added to the Figure legend.

Table explaining the changes in the figures

Main figures				
Original version		Note	Revised version	
Figure 1	A	Unchanged	Figure 1	A
	B	Unchanged		B
	C	Unchanged		C
	D	Unchanged		D
	E	Order of figures changed		G
	F			E
	G			F
	H			H
	I	Unchanged		I
	J	Unchanged		J
	K	Unchanged		K
	L	Unchanged		L
	Figure 2	A		Unchanged
B		Graphs simplified and merged	B	
C			C	
D		Unchanged	D	
E		Unchanged	E	
F		Unchanged	F	
G		Graphs merged	G	
H		Graphs merged	H	
I		Figures merged	I	
J				
K				
L		Graphs simplified and merged		
M				
Figure 3	A	Unchanged	Figure 3	A
	B	Unchanged		B
	C	Graphs merged		C
	D	Graphs merged		D
	E	Unchanged		E
	F	Figures merged		F
	G	Graphs merged		G
	H			
Figure 4	A	Unchanged	Figure 4	A
	B	Unchanged		B
	C	Unchanged		C
	D	Unchanged		D
	E	Unchanged		E
	F	Unchanged		F

	G	Unchanged		G
Figure 5	A	Unchanged	Figure 5	A
	B	Graphs simplified and merged		B
	C			C
	D	Unchanged		D
	E	Unchanged		S11A
	F	Moved to supplement		E
	G	Unchanged		S11B
	H	Moved to supplement		F
	I	Unchanged		
Figure 6	A	Unchanged	Figure 6	A
	B	Unchanged		B
	C	Removed		
	D	Unchanged		C
		Newly added figure 7	Figure 7	
Supplement figures				
Original version		Note	Revised version	
Figure S1		Unchanged	Figure S1	
Figure S2		Removed		
Figure S3		Removed		
Figure S4	A	Simplified	Figure S2	A
	B	Unchanged		B
	C	Unchanged		C
Figure S5		Unchanged	Figure S3	
Figure S6		Removed		
Figure S7		Unchanged	Figure S4	
Figure S8	A	Unchanged	Figure S5	A
	B	Graphs simplified		B
	C	Merged with E		C
	D	Merged with F		D
	E	Merged with C		C
	F	Merged with D		D
	G	Unchanged		E
	H	Unchanged		F
	I	Unchanged		G
Figure S9	A	Merged with C	Figure S6	A
	B	Merged with D		B
	C	Merged with A		A
	D	Merged with B		B
	E	Unchanged		C
	F	Unchanged		D
	G	Unchanged		E

	H	Removed		
	I	Unchanged		F
Figure S10		Unchanged	Figure S7	
Figure S11	A	Simplified	Figure S8	A
	B	Graphs simplified		B
	C	Unchanged		C
	D	Unchanged		D
	E	Simplified		E
	F	Graphs simplified		F
	G	Unchanged		G
	H	Figures merged		H
	I			
		Unchanged		
	J	Simplified		I
	K	Unchanged		J
	L	Simplified		K
	M	Simplified		L
	N	Unchanged		M
	O	Simplified		N
	P	Unchanged		O
Q	Unchanged	P		
Figure S12	A	Simplified	Figure S9	A
	B	Graphs simplified		B
	C	Unchanged		C
	D	Graphs simplified		D
Figure S13		Unchanged	Figure S10	

Dear Prof. Steingrímsson

Thank you for the submission of your revised manuscript to EMBO reports. We have now received the full set of referee reports that is copied below.

As you can see, the referees find that the study has been significantly improved during revision and recommend publication. Before I can accept the manuscript, I need you to address some minor points below:

- Please provide a 'Disclosure and competing interests statement'. For more information see <https://www.embopress.org/page/journal/14693178/authorguide#conflictsofinterest>
- Funding information in the online manuscript tracking system is currently only provided in the Comments box. Please remove this text and enter each funder and its grant separately.
- Suppl. Table 3 is a dataset and needs to be corrected to Dataset EV1 (callouts included). It needs a legend in a separate tab of the .xls file.
- You currently have 11 Supplementary figures separately uploaded. You can either bundle all Supplementary figures in an Appendix pdf or choose 5 that you wish to 'upgrade' to Expanded View figures and bundle the other ones in an Appendix. The Appendix pdf contains all figures with their legends and has a table of contents with page numbers. The nomenclature is Appendix Figure S#. Expanded View figures are called Figure EV#. Their legend is in the manuscript file in a section called "Expanded View Figure Legends". We need individual figure files for these.
- Importantly: we do not allow splitting one supplementary figure into 3 parts (Figure S8). Please transform these parts into 3 individual figures.
- If the Supplement Tables 1 and 2 need to stay in the manuscript, they need to be renamed to Table 1 and Table 2 and should be placed after the main figure legends (callouts also need to be updated).
- The manuscript sections should be in the following order: Title page - Abstract & Keywords - Introduction - Results - Discussion - Methods - Data Availability - Acknowledgments - Disclosure Statement & Competing Interests - References - Figure Legends - Tables with legends - Expanded View Figure Legends.
- Data availability section: please remove the reviewer access and add a direct and accessible link to the dataset GSE267956. It needs to be accessible once the paper is online, but we already need the direct link.
- Please remove the DOIs from the reference list.
- Source data for Figure 2H: It appears that the blots for the 'ew' and 'mi' conditions are from replicate 1, while the left-most blots for the 'Wh' condition is from replicate 2. Can you please check the composition and source data for that panel?
- Figure 3F: the blots for the delta316-326 condition appear to be from replicate 1 but the 378* and 326* blots seem to be from replicate 2. Can you please check the composition of this figure panel and the source data?
- We perform a routine analysis on all quantitative source data provided. In this case there appear to be several duplications and/or very similar values provided in the .xls files provided for Figure 3C, 3E, and 3G. The duplications are color-coded. Please carefully check the attached files, the original data and quantifications and clarify the aberrant duplications, in particular for the replicates in Figure 3C.
- Source data: please upload individual folders for each figure.
- Experimental mouse work, methods section of the manuscript: please provide information on the housing and husbandry conditions. State details of authority granting ethics approval and provide the reference number for approval. Include a statement of compliance with ethical regulations.
- Please use our 'Structured Methods' format, which is required for all research articles. According to this format, the Methods section includes a Reagents and Tools Table (listing key reagents, experimental models, software and relevant equipment and including their sources and relevant identifiers) followed by a Methods and Protocols section. More information on how to adhere to this format as well as a downloadable template (.docx) for the Reagents and Tools Table can be found in our author guidelines: <https://www.embopress.org/page/journal/14693178/authorguide#structuredmethods>.

An example of a Method paper with Structured Methods can be found here:
<https://www.embopress.org/doi/10.15252/msb.20178071>.

- Our production/data editors have asked you to clarify several points in the figure legends (see below). Please incorporate these changes in the manuscript and return the revised file with tracked changes with your final manuscript submission.

A) Please note that the legends for figures 4b-c is not provided in the sequential manner (legend for figure 4c is provided before legend of figure 4b). This needs to be rectified.

B) Please note that the exact p values are not provided in the legends of figures 2b, d, f-g, i; 3c, e, g; 4b, d, g; 5b, d-f.

C) Please note that in figures 2b, d, f-g, i; 3c, g; 4g; 5b, d, there is a mismatch between the annotated p values in the figure legend and the annotated p values in the figure file that should be corrected.

D) Please note that scale bar and its definition are missing for figure 1k.

- I introduced some minor changes to the Abstract. Please find my suggestions at the end of this e-mail.

- Finally, EMBO Reports papers are accompanied online by

A) a short (1-2 sentences) summary of the findings and their significance,

B) 2-3 bullet points highlighting key results and

C) a schematic summary figure that provides a sketch of the major findings (not a data image).

Please provide the summary figure as a separate file in PNG or JPG format at a size of 550x300-600 pixels (width x height).

Please note that the size is rather small and that text needs to be readable at the final size. Please send us this information along with the revised manuscript.

- On a different note, I would like to alert you that EMBO Press offers a new format for a video-synopsis of work published with us, which essentially is a short, author-generated film explaining the core findings in hand drawings, and, as we believe, can be very useful to increase visibility of the work. This has proven to offer a nice opportunity for exposure i.p. for the first author(s) of the study. Please see the following link for representative examples and their integration into the article web page:

<https://www.embopress.org/doi/full/10.15252/embj.2019103932>

Referee #1:

As the authors altered their manuscript to focusing on the structural aspects of the new mutation and not exploring the MITF mutation in context to the melanoma phenotype my initial concerns have been adequately answered. I believe the resulting edited manuscript is much more focused and have no more concerns.

Referee #2:

Vu HN, et al. have addressed all of this reviewers comments and am therefore recommending publication.

Abstract

MITF, a basic-Helix-Loop-Helix Zipper (bHLHZip) transcription factor, plays vital roles in melanocyte development and functions as an oncogene. We perform a genetic screen for suppressors of the Mitf-associated pigmentation phenotype in mice and identify an intragenic Mitf mutation that terminates MITF at the K316 SUMOylation site, leading to loss of the C-end intrinsically

disordered region (IDR). The resulting protein is more nuclear but less stable than wild-type MITF and retains DNA-binding ability. As a dimer, it can translocate wild-type and mutant MITF partners into the nucleus, improving its own stability thus ensuring nuclear MITF supply. smFRET analysis shows interactions between K316 SUMOylation and S409 phosphorylation sites across monomers; these interactions largely explain the observed effects. The recurrent melanoma-associated E318K mutation in MITF, which affects K316 SUMOylation, also alters protein regulation in concert with S409. This suggests that residues K316 and S409 of MITF are impacted by SUMOylation and phosphorylation, respectively, mediating effects on nuclear localization and stability through conformational changes. Our work provides a novel mechanism of genetic suppression, and an example of how apparently deleterious mutations lead to normal phenotypes

All editorial and formatting issues were resolved by the authors.

Prof. Eiríkur Steingrímsson
Univeristy of Iceland
Department of biochemistry and molecular biology; Faculty of medicine
Reykjavík 101
Iceland

Dear Eiríkur,

Thank you for clarifying the few remaining editorial issues. I am now very pleased to accept your manuscript for publication in the next available issue of EMBO reports. Thank you for your contribution to our journal.

Kind regards,

Martina
